# Balancing the efficacy and safety of chimeric antigen receptor T-cell therapy by affinity combination

Linda Warmuth [1,2,13], Sarah Dötsch [1,13], Manuel Trebo [1], Sarah Bellucci[1], Sophia Engels[1], Rafael Valdivia Manrique[1], Karl Moukarzel[1,2], Julius M. Schütz[1,2], Monika Hammel[1], Adrian Straub [1], Sabrina Wagner [1,2], Anna Hochholzer[1], Ciro Salinno[1], Jacqueline Seigner [3], Charlotte U. Zajc [3,4], Georg P. Schmidt[5], Josefine Michael[6], Thomas Nerreter [6], Michael Hudecek [6,7,8,9], Michael W. Traxlmayr [3], Monica Casucci [10], Stanley R. Riddell [11], Mateusz P. Poltorak[1], Dirk H. Busch [1,12,14] ✉ & Elvira D'Ippolito [1,12,14] ✉

Recent studies suggest that Chimeric Antigen Receptor (CAR) binding affinity to its ligand affects CAR-T-cell functionality. Affinity engineering towards lower binding strengths might mitigate therapeutic side effects arising from intense CAR-T-cell activation as well as tumor relapse due to antigen-escape or limited persistence of CAR-T cells during sustained activation via high-affinity receptors. Here we characterize a broad range of CARs with varying affinities to the same target epitope and leverage the insights we gain to design a combined high- and low-affinity CAR product. While CAR affinity impacts in vitro functionality minimally, it strongly correlates with tumor control in vivo. Low-affinity binders cause only mild cytokine release syndrome (CRS) in humanized mouse models at the expense of anti-tumour efficiency. In mixtures with low-affinity CARs, high-affinity CARs maintain strong functionality while showing reduced signs of exhaustion and monocyte-induced cytokine production, compared to high-affinity CAR-T cells alone. In long term in vitro and in vivo settings, low-affinity CAR-T cells dominate over time, proving more resilient to chronic antigen exposure. Overall, our findings demonstrate that affinity combination represents a promising strategy to generate more effective CAR-T-cell products with an improved therapeutic index, beyond affinity engineering alone.

Adoptive cell therapy (ACT) with patient-derived T cells engineered with tumor-specific transgenic receptors emerged as a beacon of hope in the cancer field. T cells equipped with chimeric antigen receptors (CAR) have demonstrated remarkable success, in particular in refractory and relapsed B-cell malignancies expressing CD19 or BCMA[1–6], leading to the FDA approval of currently seven CAR-T-cell products. Nevertheless, durable responses remain limited, motivating ongoing clinical investigation of novel approaches, including the use of fully human single-chain variable fragments (scFv) to minimize anti-CAR immune responses directed against murine scFvs[7]. The success of these therapies is now being extended to severe autoimmune indications with very promising first clinical data[8,9].

However, initial therapeutic efficacy comes at the cost of acute side effects, namely cytokine release syndrome (CRS) and immune

effector cell-associated neurotoxicity syndrome (ICANS). While CRS and ICANS are usually reversible, they occur frequently[10–14] and must be managed accordingly to circumvent complications[15,16]. Even though CAR-T-cell activation represents the initial trigger, a broader part of the immune system is engaged, and monocyte-derived interleukin 6 (IL-6) and interleukin 1 (IL-1) are the main drivers of CRS pathophysiology[17,18]. Moreover, recent studies have demonstrated a correlation between the severity of CRS and the extent of hematotoxicity over-time[19]. Notably, prolonged cytopenia may persist beyond the resolution of CRS, posing a significant risk of infection and an increased likelihood of hemorrhagic complications.

Additionally, the effectiveness and wider use of current CAR-T-cell products are limited due to the common occurrence of disease relapses in patients[4,20,21]. Being derived from antibodies, the affinities of scFvs selected for CAR-T-cell generation[22] are typically 100 to 1000-fold higher than the physiological binding strength of natural T-cell receptors (TCR) towards their cognate antigens (CAR $K_D$: $10^{-6}$-$10^{-9}$ M versus TCR $K_D$: 1–200 μM)[23,24]. The resulting 'supra-physiological' T-cell activation via high-affinity antigen binding is associated with accelerated exhaustion and reduced persistence of the transferred cells, as described for nearly all commercially available products[25–27]. High CAR binding affinity can also enhance the selection towards antigen loss or downregulation variants[28]. Besides these more commonly observed immune evasion mechanisms, recent studies also highlight the role of transient and short-term antigen escape, protein transfer and antigen depletion for the occurrence of disease relapses[29]. For example, trogocytosis—an intercellular contact-dependent material transfer—has lately been described as a mechanism interfering with CAR-T-cell efficacy. In particular, high-affinity CAR-T cells can exert strong trogocytosis, which can alter tumor recognition by reducing antigen density; in addition, antigen transfer to the CAR-T cells themselves can confer susceptibility to fratricide and affect their in vivo persistence[28,30]. These limitations spark an urgent need for the design of CAR-T-cell products with more effective properties.

Hence, multiple alternatives to commercially used CAR engineering approaches have been developed with the focus on bringing the synthetic receptor closer to physiological TCR characteristics[31–34]. These approaches include adjustments of the CAR antigen-binding affinity towards significantly lower affinities. For conventional T-cell responses, it is well established that T-cell functionality can be differentially affected by receptor affinity. Acute protective immunity is usually mediated by the recruitment of highly polyclonal T-cell populations with a broad spectrum of TCR-target antigen affinities. Out of these, T cells possessing higher TCR affinities for cognate epitope: major histocompatibility complex (MHC) ligands are more strongly expanded and confer superior in vivo target cell recognition and efficacy. Too high TCR affinities—beyond a not very well-defined threshold—do not further improve efficacy and are less dominant in vivo[35–37]. During chronic antigen exposure, high-affinity T cells are more prone to exhaustion and immune senescence. Therefore, T cells with lower-affinity TCRs can become more dominant and potentially also more relevant for long-term protection[38–41]. In line with these observations from normal T-cell responses, recent studies with lower-affinity CAR-T cells demonstrated comparable or even augmented cytotoxic activity[42,43], while developing milder side effects[25]. These data have received increasing attention, since they indicate that optimal receptor binding affinity design could be used for both the reduction of side effects as well as the improvement of CAR-T-cell function and maintenance. However, determining an optimal affinity remains a demanding task, potentially requiring case-by-case optimization for each CAR:antigen pair as well as further adjustment to tumor characteristics (e.g., tumor burden and heterogeneity).

This study systematically examines how variations in CAR affinity influence the functional efficacy and safety profile of CAR-T cells and, more importantly, how integrating receptors of differing affinities can produce CAR-T-cell products with a more favorable therapeutic balance. We generate and comprehensively characterize CAR-T cells spanning a wide range of affinities down to TCR-like binding strengths and observe that while in vitro activation and cytotoxicity remain largely unaffected, in vivo anti-tumor efficacy strongly correlates with CAR affinity. Notably, low-affinity CAR-T cells substantially attenuate treatment-associated toxicity, and a combination of high- and low-affinity receptors enhances efficacy while limiting exhaustion and adverse events commonly associated with single-receptor high-affinity CAR-T-cell products.

Overall, our work highlights affinity combination as a practical approach to enhance CAR-T-cell efficacy while maintaining safety, paving the way for more balanced and widely applicable next-generation therapies.

## Results

### In vitro characterization of high- and low-affinity CAR-T cells

In order to systematically explore the effect of receptor affinity on CAR-T-cell characteristics, we selected two anti-human CD19 scFvs: JCAR017, which is derived from the high-affinity murine FMC63 antibody and is the central component of most commercially approved anti-CD19 CAR-T-cell products, and JCAR021, which is a fully humanized anti-CD19 CAR. We also employed the anti-CD19 scFv of the recently published CAT CAR as a reference for an affinity-reduced receptor with enhanced functionality[25]. To determine affinity, all scFvs were expressed as recombinant proteins conjugated to a Twin-Strep-tag (Fig. 1a), enabling their immobilization on a StrepTactin-coated surface for surface plasmon resonance (SPR) measurements. Using an engineered stable version of the extracellular domain (ECD) of the recombinant CD19 protein[44], we confirmed the high affinity of JCAR017 ($K_D = 2.7 \times 10^{-9}$ M). In comparison, JCAR021 showed a reduced affinity ($K_D = 1.4 \times 10^{-7}$ M) closer to, but still slightly lower than the published CAT scFv ($K_D = 5.5 \times 10^{-8}$ M) (Figs. 1b and S1a)[25]. Moreover, we observed that the differences in $K_D$ values were primarily driven by the dissociation kinetics ($k_{off}$), rather than the association rate ($k_{on}$) (Fig. 1b), consistent with other reports[25]. The epitope position within the three-dimensional structure of an antigen can affect CAR accessibility and, consequently, the functions of engineered T cells. Notably, our chosen scFvs recognize the same conformational epitope of CD19. This is evidenced by the lack of ECD CD19 binding to immobilized low-affinity JCAR021 in the presence of the high-affinity JCAR017 scFv (Fig. 1c, d).

To assess the impact of receptor affinity on T-cell functionality, we introduced the scFvs into a second-generation CAR backbone. The scFvs were linked via a CD28 transmembrane domain to the intracellular signaling domains CD3ζ and 4-1BB. An inert truncated epidermal growth factor receptor (EGFRt) was incorporated into the construct and utilized to track the efficiency of T-cell engineering[45–47], while a triple repeat sequence of Strep-tag II (STII)[46,48] was attached to the extracellular hinge domain to verify surface expression of the CAR construct (Fig. 1a). The vectors could be efficiently delivered with stable CAR surface expression (Fig. 1e). To examine how CAR affinity affects T-cell activation, we introduced the various CARs into a Nur77-tdTomato-Jurkat reporter cell line[49]. Within this reporter system, the fluorescent protein tdTomato is co-expressed with the early gene of TCR signaling Nur77, thus enabling a fast determination of CAR-T-cell sensitivity after antigen encounter[50]. Receptor affinity did not impact the activation strength (functional avidity) in this in vitro assay (Fig. 1f). We then introduced the described CARs into primary human T cells and observed that both high- and low-affinity CAR-T cells showed similar functionality in terms of cytokine production and cytotoxicity when co-cultured with CD19-expressing tumor cells (Figs. 1g, h and S1b, c), even at low effector-to target ratios and with CD19-low-expressing target cells (Fig. S1d, e).

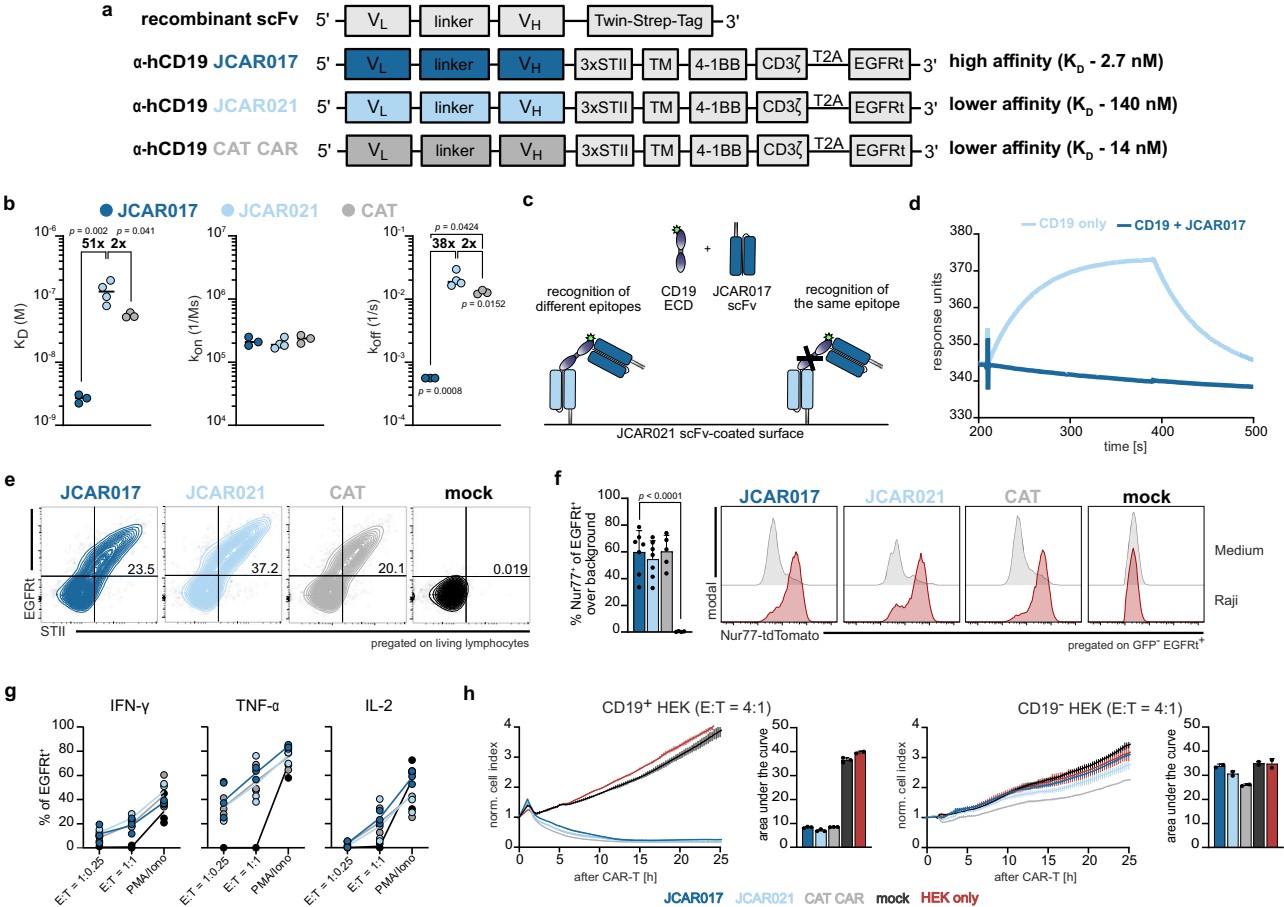

**Fig. 1 | In vitro characterization of high- and low-affinity CAR-T cells.**
**a** Schematic of the scFv design for periplasmic expression in *E. coli* and CAR construct structures for retroviral gene delivery. **b** Surface plasmon resonance (SPR) analysis of scFv binding kinetics, including dissociation constant ($K_D$), association rate ($k_{on}$) and dissociation rate ($k_{off}$). Each dot represents an individual replicate ($n = 3$; JCAR021 $n = 4$, dash indicates the mean measured with the capture of 50 nM scFv for 60 s with a 10 μL/min flow rate). **c** Schematic representation of epitope mapping to determine similarities in the recognition sites of JCAR017 and JCAR021 scFvs. **d** SPR sensorgram showing soluble CD19 binding to JCAR021 scFv in the presence or absence of JCAR017 scFv. **e** Representative flow cytometry plots of transduction efficiency (EGFRt⁺) and surface expression level (STII⁺) of JCAR017, JCAR021 and reference CAT CAR in primary human T cells. **f** Quantification of Nur77-tdTomato expression in CAR-transduced Jurkat cells stimulated with CD19⁺ GFP⁺ Raji tumor cells at an effector-to-target (E:T) ratio of 1:1 or cultured in medium,

background-subtracted. Each dot represents the mean of technical triplicates. Bars indicate the mean + SD across biological replicates ($n = 7$; CAT $n = 5$). Additionally, representative histograms displaying Nur77-tdTomato signal are shown.
**g** Intracellular cytokine production quantified after 5 h co-culture with CD19⁺GFP⁺ Raji cells at the indicated E:T ratios. Each dot represents the mean of technical triplicates of one independent biological replicate ($n = 3$). PMA/ionomycin was used as a positive control. **h** Representative xCelligence impedance-based killing curves with CD19⁺ (left) or wildtype (WT) (right) HEK cells, including quantification by the area under the curve (AUC), are shown. Each dot represents a technical replicate of one out of three independent biological experiments. Data is presented as mean + SD ($n = 3$). Statistical analyses were performed for (**b**, **f**) using one-way ANOVA for multiple comparisons between the individual scFvs (**b**) or with JCAR017 as reference (**f**). \*$p < 0.05$, \*\*$p < 0.01$, \*\*\*$p ≤ 0.001$, \*\*\*\*$p ≤ 0.0001$.

---

Overall, we did not observe any significant impact on the in vitro functionality of CAR-T cells even when we reduced the CAR affinity by 50-fold.

**Ligand binding affinity has only scarce impact on in vitro functionality of CAR-T cells**
As the above investigated affinity range left in vitro functionality apparently unaffected but was still an order of magnitude higher than physiological TCR binding strengths, we explored the possibility of reducing the range of CAR affinities even down to physiological TCR:pMHC interactions. We used educated exchanges of 32 individual residues into the low-affinity binder JCAR021, which harbored a high probability to reduce the scFv affinity without compromising epitope specificity (Fig. 2a and Tab. S1). As the scFv dissociation kinetics ($k_{off}$ rates) were the main determinant of the overall $K_D$ in our previous measurements (Fig. 1b), we adapted our flow cytometry-based TCR:ligand $k_{off}$-rate technique[51–53] to screen the library of mutated

scFvs for potential changes in dissociation kinetics. Besides increased throughput compared to SPR, this method enables the monitoring of monomeric ligand dissociations under most physiological conditions (e.g., cell surface distribution and/or endogenous expression of the respective target antigen). For this purpose, mutated scFvs were double tagged in frame with a Strep-tag and a Tub-tag sequence (scFv Flexamer) (Fig. 2a). The tags enable the multimerization of monomeric scFvs on a StrepTactin backbone (scFv *Strep*Tamer) and the site-specific conjugation of a fluorescent dye[54] (Fig. 2b). scFv *Strep*Tamers were used to stain PBMC-derived target CD19⁺ B cells. The administration of D-biotin leads to the displacement of the StrepTactin backbone due to its higher affinity compared to the Strep-tag, leaving monomeric scFvs attached to the target cells. The monomeric scFvs dissociate from cognate epitopes according to their intrinsic affinity, and the decay of the fluorescent signal can be translated into the $k_{off}$-rate (Figs. 2b and S2a). The individual labeling of B cells with distinct fluorescence-conjugated anti-CD45 antibodies enabled multiplexed

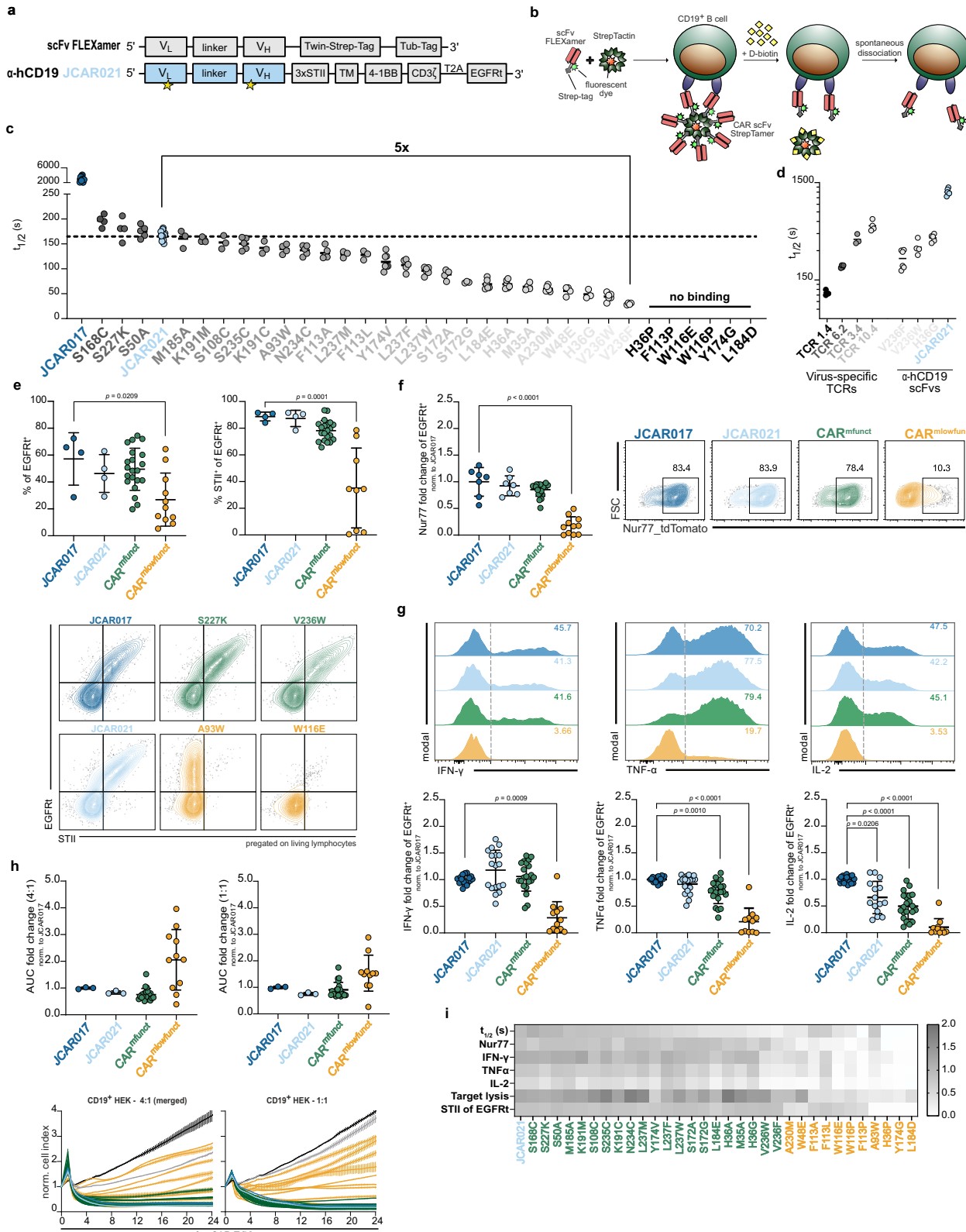

k$_{off}$-rate measurements of up to 16 different scFvs simultaneously (Fig. S2b). In line with the SPR measurements, we observed a much slower dissociation kinetics for the JCAR017 scFv in comparison to the JCAR021 scFv. Quantitatively, JCAR017 and JCAR021 scFvs possessed k$_{off}$-rates in a similar range as measured via SPR (JCAR017: SPR $k_{off} = 5.6 \times 10^{-4} s^{-1}$ vs flow cytometry $k_{off} = 2.3 \times 10^{-4} s^{-1}$; JCAR021: SPR $k_{off} = 2.1 \times 10^{-2} s^{-1}$ vs flow cytometry $k_{off} = 4.1 \times 10^{-2} s^{-1}$), thus validating

the reliability of the flow cytometry-based measurement of scFv dissociation kinetics. For 26 JCAR021 scFv mutants, we were able to determine robust and highly reproducible k$_{off}$-rate values and these demonstrated coverage of a broad range of distinct dissociation kinetics for individual amino acid exchanges (Figs. 2c and S2c). The remaining six mutants showed either an overall loss (H36P, F113P, W116E, Y174G) or largely reduced and variable (W116P, L184D) CD19

**Fig. 2 | Ligand binding affinity has only scarce impact on in vitro functionality of CAR-T cells. a** Schematic of the scFv sequence used to generate fluorescently labeled scFv FLEXamers along with the CAR sequence for retroviral transduction to construct the affinity library derived from the JCAR021 framework. Mutagenesis regions are indicated by stars. **b** Principle of the flow cytometry-based assay to assess monomeric scFv:epitope dissociations. **c** Quantification of scFv dissociation kinetics ($k_{off}$-rate) by one-phase exponential decay curve fitting, expressed as dissociation half-life ($t_{1/2}$) ($n = 3$, 30 min 20 °C). **d** Comparative analysis of TCR and scFv koff-rates ($n \geq 3$, 30 min 4 °C). **e** Transduction efficiency (EGFRt⁺) (upper left), CAR-expression (STII⁺) (upper right) and representative flow cytometry plots (below) in primary human T cells ($n = 3$). **f** Fold change in Nur77-tdTomato expression of CAR-transduced Jurkat cells upon stimulation with CD19⁺GFP⁺ Raji cells (E:T 1:1) or incubation in medium for 3 h ($n \geq 5$). Representative flow cytometry plots are shown on the right. **g** Intracellular cytokine production in CAR-engineered primary human T cells following 5 h co-culture with CD19⁺GFP⁺ Raji cells at an E:T of 4:1 ($n \geq 3$) with representative flow cytometry histograms. **h** Impedance-based xCelligence killing curves (below) and quantification of CAR-T-cell killing (upper) as area under the curve (AUC) ($n \geq 2$). **i** Heatmap summarizing the functional properties of JCAR021, CAR^mfunct and CAR^mlowfunct in vitro. **f–i** Data are normalized to the average value of JCAR017. Data are expressed as mean ± SD. Each dot represents the mean of technical replicates for JCAR017 and JCAR021 and the mean of an independent experiment for the JCAR021 mutants. Statistical analyses were performed in (**e–h**) using a one-way ANOVA test for multiple comparisons with JCAR017 as reference. *$p < 0.05$, **$p < 0.01$, ***$p \leq 0.001$, ****$p \leq 0.0001$.

ligand binding (Figs. 2c and S2d), which precluded robust $k_{off}$-rate measurements. By comparing the binding strength landscape of virus-specific TCRs[55] with the JCAR021 mutants (these experiments had to be performed at lower temperatures (4 °C) to allow measurements of fast TCR:pMHC dissociation kinetics), we could demonstrate that some JCAR021 mutants with the fastest $k_{off}$-rates fell indeed within the range of TCR affinities (Fig. 2d). Notably, the temperature did not qualitatively alter the dissociation kinetics of either interaction (Fig. S2e).

We introduced the JCAR021 mutant scFvs into the same second-generation backbone as before (Fig. 2a). As stable receptor surface expression is one determining parameter for target recognition and CAR-T-cell functionality, we evaluated the efficacy of vector delivery and CAR surface expression via the transduction marker EGFRt and the CAR expression marker STII. Most of the mutants preserved an optimal CAR surface expression, as indicated by the correlation between STII and EGFRt, highly comparable to JCAR017 and JCAR021 (21/32, 66%, referred to as CAR^mfunct indicated in green). In contrast, the transgenic receptor was barely detectable at the cell surface of the remaining CAR-T cells (11/32, 34%, referred to as CAR^mlowfunct indicated in yellow), for which we observed either weak co-expression of the CAR and EGFRt markers, or transduction rates below 20% (Figs. 2e and S3a). Of note, the two CAR variants with EGFRt/STII co-expression levels apparently comparable to those of the CAR^mfunct group—namely F113L and W116E—exhibited little CAR surface expression.

Next, we re-expressed the JCAR021 mutant library into the Nur77-tdTomato-Jurkat reporter cell line[49]. The functional activation pattern among T cells engineered with JCAR017, JCAR021, and CAR^mfunct mutants was very similar despite the strong variations in antigen-binding affinity. In contrast, CAR^mlowfunct-T cells showed a drastically reduced activation pattern (Figs. 2f and S3b). Furthermore, when re-expressed in primary human T cells, CAR^mfunct-engineered T cells closely resembled cytokine production (IFN-γ, TNF-α and IL-2) and cytotoxicity as compared to the high-affinity binder JCAR017 or the original JCAR021. CAR^mlowfunct-engineered T cells, instead, showed a reduction in cytokine production or did not respond to the CD19 stimulus at all. Moreover, these T cells were predominantly unable to control target cell growth at low effector to target ratios (Figs. 2g, h and S3c, d). More challenging effector-to-target ratios and the use of CD19-high- and low-expressing target tumor cell lines did not improve resolution with respect to the dependence of in vitro target cell killing on CAR affinity, except for the ultra-low, TCR-like-affinity CAR mutants (Fig. S4). Furthermore, we investigated the extent of interaction with the target cell membrane (trogocytosis) of selected mutants in comparison to JCAR017 and JCAR021 (Fig. S5a). As expected, the extent of trogocytosis decreased along with CAR ligand binding affinity (Fig. S5b, c), in line with recently published studies[28,29].

In summary, surface expression of the CAR appears to have a greater impact on measurable in vitro functionality than the binding affinity of the scFv (Fig. 2i). Among the stable CAR constructs within the CAR^mfunct group, functional differences were observed only at ultra-low affinity levels—within the physiological range of functional TCRs

(e.g. V236W and V236F). These variants showed slight reductions in cytokine production, notable differences in trogocytosis and reduced target cell killing at low effector-to-target ratios and low antigen density. Overall, these findings demonstrate that the threshold of CAR binding strength that still translates into effective in vitro functionality is remarkably low, even reaching down to low-affinity interactions with extremely fast $k_{off}$-rates ($t_{1/2} = 50$ s at 20 °C).

## CAR ligand binding affinity influences recruitment, activation and anti-tumor efficacy in vivo

Within CARs with preserved and very similar surface expression, we observed comparable in vitro functionality irrespective of antigen-binding affinities, except when we analyzed CAR-ligand interactions as low as TCR-like binding affinities. Next, we assessed whether these similarities in functionality were also preserved in vivo. For this purpose, we selected the JCAR017 and JCAR021 CARs alongside three additional mutants that cover a broad affinity spectrum. The mutant L237W ($t_{1/2} = 96$ s) exhibited lower affinity than JCAR021 ($t_{1/2} = 167$ s) despite preserved in vitro functionality. We detected mutant M35A to be positioned at the threshold of in vitro functional CARs ($t_{1/2} = 64$ s) with further reduced affinity. Finally, mutant V236W harbored one of the fastest $k_{off}$-rates ($t_{1/2} = 44$ s) and displayed reduced functionality compared to other CAR^mfunct even in vitro. Notably, the three selected mutants belonged to the CAR^mfunct group with stable surface expression, therefore attributing potential differences in CAR performance primarily to the scFv affinity.

We initially assessed in vivo functionality using an immunocompromised mouse model with implanted CD19⁺ Raji lymphoma cells. (Fig. 3a). These cells were additionally modified to express firefly luciferase for bioluminescence imaging and green fluorescent protein (GFP) for peripheral detection. The T-cell products of the five CAR constructs demonstrated comparable levels of CAR expression with a strong correlation between EGFRt and STII (Figs. 3b and S6a), an equal distribution of CD8⁺ and CD8⁻ T cells (Fig. 3c) and a similar ex vivo phenotype (Fig. 3d) prior to infusion. Eleven days after T-cell transfer, the majority of mice treated with non-transduced T cells (mock) had to be sacrificed due to rapid tumor progression, while the ultra-low-affinity mutant V236W provided minor survival benefits. Mice treated with the low-affinity CARs JCAR021, L237W and M35A demonstrated improved survival. However, in contrast to our results observed in vitro, lower-affinity CARs were less effective in controlling tumor progression as the high-affinity JCAR017-T cells (Figs. 3e, f and S6b).

To gain a clearer understanding of the factors contributing to the reduced in vivo efficacy of lower-affinity CARs, we examined the proliferative capacity and activation of transferred cells. JCAR017-T cells were significantly stronger expanded in peripheral blood as early as seven days post-transfer and continued to expand throughout the observation period. In contrast, the lower-affinity CARs reached higher numbers compared to JCAR017 in peripheral blood at day 14 post CAR-T-cell transfer. The only exception was the ultra-low mutant V236W-treated cohort, in which CAR-T cells were detectable at an early time

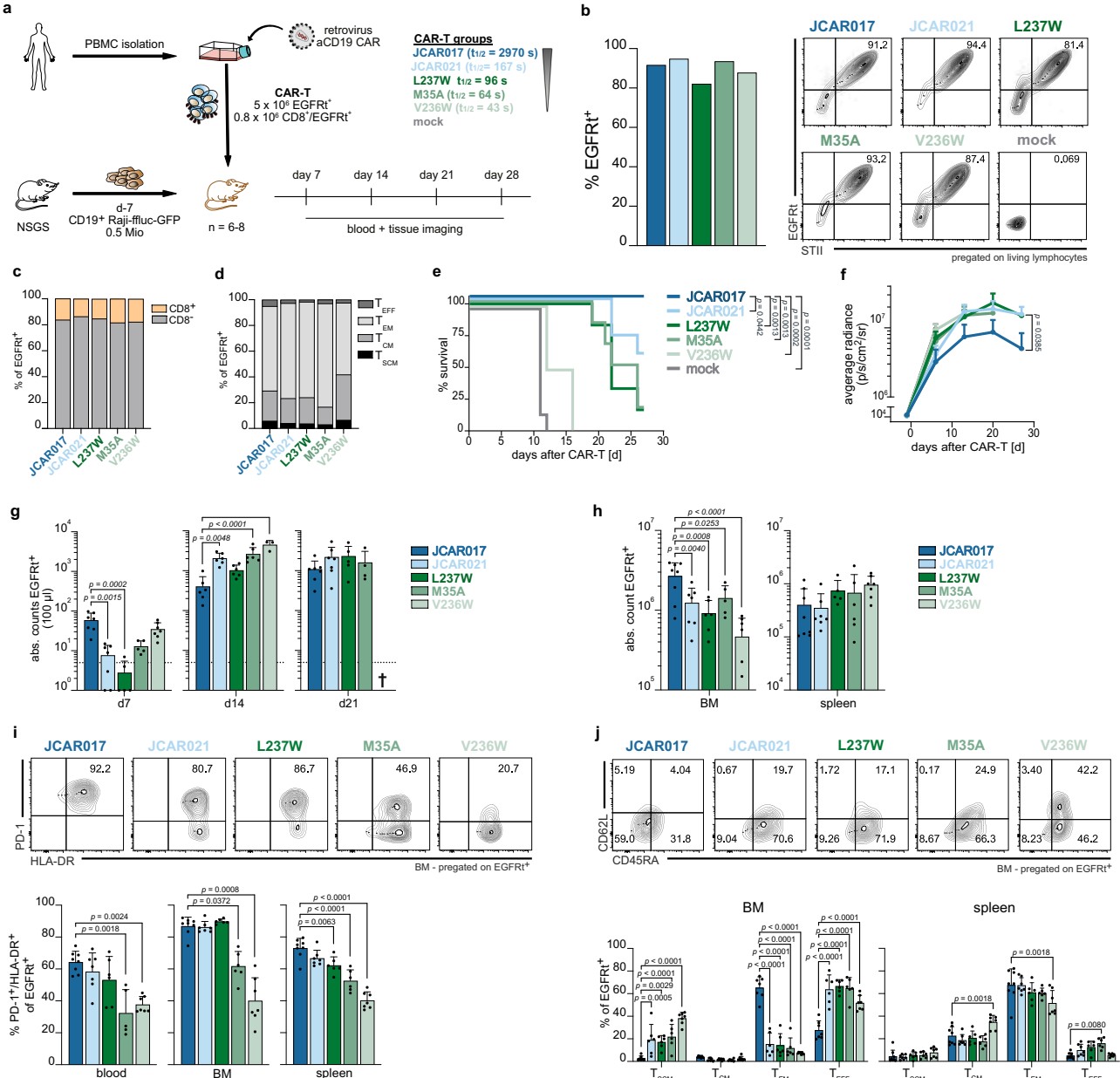

**Fig. 3 | CAR ligand binding affinity influences recruitment, activation, and antitumor efficacy in vivo. a** Schematic overview of the experimental setup in a xenograft tumor model (Raji ffluc) employing bulk CAR-T cells. **b** Quantification of the transduction efficiency (EGFRt⁺) (left) and flow cytometry plots depicting co-expression of transduction marker (EGFRt⁺) and CAR surface expression (STII⁺) of CAR-transduced primary human T cells pre-infusion. **c** T-cell subset distribution and **d** phenotypic analysis of the infused CAR-T-cell products (T_scm CD62L⁺CD45RA⁺; T_cm CD62L⁺CD45RA⁻; T_em CD62L⁻CD45RA⁻; T_eff CD62L⁻CD45RA⁺). **e** Kaplan–Meier survival curve for mice treated with respective CAR-T-cell products. **f** Residual tumor burden quantified as the maximum photon per second per cm² per steradian depicted as median + SD. **g** Quantification of CAR-T-cell expansion in the circulation (EGFRt⁺ cells) assessed in blood at the indicated time points post transfer. Each dot represents an individual mouse. **h** CAR-T-cell distribution at sacrifice displayed as absolute cell numbers in bone marrow (BM) and spleen. Each dot represents an individual mouse. **i** Representative flow cytometry plots and quantification of PD-1 and HLA-DR co-expression in EGFRt⁺ CAR-T cells in blood, BM and spleen at sacrifice. Each dot represents an individual mouse. **j** Phenotypic distribution of EGFRt⁺ CAR-T cells in BM and spleen. Each dot represents an individual mouse. Data are representative of two independent experiments derived from two different donors. Data are expressed as mean + SD (JCAR017 $n = 8$; JCAR021 and M35A $n = 7$; L237W and V236W $n = 6$). Statistical testing by Log-rank test (**e**) one-way ANOVA (**g**–**j**), or two-way ANOVA (**f**) using JCAR017 as reference, *$p < 0.05$, **$p < 0.01$, ***$p \leq 0.001$, ****$p \leq 0.0001$.

point, similar to the JCAR017 cohort (Fig. 3g). Nevertheless, the quantity of circulating T cells does not necessarily reflect their impact on combating the disease, so we further investigated the CAR-T presence at the tumor site. Recent studies have shown that low-functional T cells can be enriched in the peripheral circulation rather than at the tumor site in cancer patients[39]. Therefore, we assessed T-cell infiltration into the bone marrow, the primary site of Raji cell localization, and

the spleen, which is only minimally invaded by tumor cells unless there is an extremely high burden (Fig. S6c). Although CAR-T cells from all groups successfully engrafted, we observed significantly higher numbers of EGFRt⁺ cells in the bone marrow of mice treated with high-affinity JCAR017-T cells (Fig. 3h). T-cell infiltration was generally similar for the low-affinity CAR-T cells, except for the ultra-low-affinity clone V236W, which showed lower abundance. Conversely, the CAR-T-cell

quantity in the spleen was equivalent in all treated mice (Fig. 3h), suggesting a preferential spatial distribution of T cells within the tumor based on receptor affinity.

Additionally, we observed a correlation between the activation status of CAR-T cells and their receptor affinity in both blood and tissues. Specifically, the frequency of activated PD-1+HLA-DR+ cells was significantly reduced in both compartments of M35A and V236W-treated mice compared to the high-affinity JCAR017 (Fig. 3i). Furthermore, JCAR017-T cells maintained a higher frequency of memory cells (CD45RA−), whereas lower-affinity CARs displayed a more differentiated phenotype (CD45RA+ CD62L−), likely due to the increasing tumor burden. Notably, the ultra-low-affinity V236W CAR-T cells retained a more naïve/stem-like phenotype, indicating less activation or recruitment into the tumor response. We did not observe these variabilities within the spleen, where we identified predominantly central or effector memory differentiated populations among the EGFRt+ CAR-T cells (Fig. 3j).

Finally, we investigated whether escalating CAR-T-cell doses could rescue the in vivo anti-tumor efficacy of low-affinity CAR-T cells (Fig. S7a, b). Low doses of JCAR017-CAR-T cells were sufficient to ensure survival of all treated mice, even though residual tumor burden correlated with CAR-T-cell doses and expansion (Fig. S7c–g). Increasing doses of low-affinity CARs increased survival rates but resulted only in a dose-dependent delay of tumor progression despite the high levels of circulating CAR-T cells (Fig. S7c–g).

Overall, we revealed that scFv affinity plays a pivotal role in determining effective in vivo anti-tumor activity by driving proper CAR-T-cell activation and recruitment to the tumor site. Large numbers of low-affinity CAR-T cells only delayed tumor progression in a dose-dependent manner.

### CAR-T-cell-associated toxicities are influenced by T-cell dose and receptor affinity

In order to investigate whether binding affinity also affects CAR-T-cell-related toxicities, we reconstituted a human-like immune compartment in immunocompromised triple transgenic NSG-SGM3 (NSG-S) mice with cord blood-derived CD34+ hematopoietic stem cells (HSCs) as described by Norelli et al.[18]. The transgenic expression of human stem cell factor, granulocyte/macrophage colony-stimulating factor (GM-CSF), and IL-3 supports the maintenance and differentiation of HSCs, especially within the myeloid cell lineages, which are key mediators for CRS[18]. We recently improved this model by reducing CD34+ HSC doses to produce cohorts of humanized mice derived from a single cord blood donor, thereby improving reproducibility and reliability of humanization[56].

To recapitulate the clinical situation, we established a system that allows the use of CD19+ tumor cell lines (Raji or Nalm-6) and healthy donor-derived PBMCs as CAR-T-cell source (Fig. 4a). We observed successful humanization within eight weeks after CD34+ HSC engraftment, indicated by the CD45+ chimerism in the blood of treated NSG-S mice. During the humanization process all model relevant human immune compartments were reconstituted—including hCD19+ B cells, hCD33+ macrophages and hCD14+ monocytes (Figs. 4b, c and S8a, b). Firstly, we assessed the feasibility of the model in inducing CRS by infusing escalating doses of JCAR017-T cells into humanized tumor-bearing NSG-S mice (hNSG-S). As expected, the anti-tumor activity and peripheral CAR-T-cell expansion correlated with CAR-T-cell doses (Fig. S8c–i). Importantly, CAR-T-cell-treated mice exhibited clinical symptoms of CRS, including weight loss and elevated serum cytokine levels (Fig. S8j, k), which were proportional to the administered T-cell dose. Symptoms were reversible in most cases, except in a certain proportion of mice (33%) treated with the highest T-cell dose (Fig. S8l).

Next, we evaluated the side effects induced by the low-affinity JCAR021. Therefore, JCAR017 and JCAR021 CAR-T cells with comparable characteristics in terms of receptor expression as well as CD8+/

CD8− distribution were transferred into tumor-bearing hNSG-S mice (Fig. S9a–c). All CAR-T-cell-treated mice developed clinical symptoms of CRS (Fig. S9d–e). Nevertheless, mice treated with JCAR021-T cells experienced a significantly milder extent of weight loss, which also peaked at earlier time points compared to JCAR017-T-cell-treated mice, thereby allowing a faster recovery (Fig. S9d). Additionally, elevated levels of circulating INF-γ, IL-10, and IL-6 cytokines were observed in both treated groups, but at lower levels in JCAR021-T-cell-treated mice (Fig. S9e). The observed pattern of CRS development was in line with the differences in in vivo functionality for the two CAR-T-cell products. Indeed, consistent with the non-humanized NSG-S model, JCAR017-T cells exhibited superior antitumor activity, greater expansion potential and faster systemic depletion of CD19+ B cells than JCAR021-T cells in the acute phase 7 days post injection (Fig. S9f–l). This more robust expansion may account for, and potentially drive, the severity of the observed toxicity.

Next, we wanted to assess the dose effect of JCAR021-T cells when compared with JCAR017. To this end, we administered high- and low-doses of the two CAR-T cells into hNSG-S mice applying the same experimental procedure as previously described (Fig. 4d, e). All CAR-T-cell-treated mice developed clinical signs of CRS, with variability depending on both the receptor affinity and T-cell dose (Fig. 4f, g). JCAR017-T cells displayed a dose-dependent increase in CRS severity. In contrast, low-affinity JCAR021-T cells treated mice displayed only intermediate CRS severity irrespective of the applied dose levels (Fig. 4f). Similarly, when we analyzed serum cytokine levels, we observed a clear dose-dependent effect for the high-affinity CAR but not for the lower-affinity CAR-T treated mice (Fig. 4g). Regarding anti-tumor effects, both doses of JCAR017-T cells successfully eradicated the tumor, with faster response rates at the highest dose, while low-affinity CAR-T cells were less effective. Nevertheless, a high dose did provide a noticeable benefit compared to untreated mice, as observed before (Fig. 4h, i). We identified that CRS development and cytokine production aligned with the degree of T-cell expansion at early time points (day five). JCAR017-T cells displayed the strongest expansion in a dose-dependent manner, followed by the high- and low-doses of JCAR021-T cells (Fig. 4j).

In summary, lowering the binding strength of the CAR can reduce the severity of CRS, although at the cost of functionality. Remarkably, CAR-T-cell dose seems to correlate with the toxicity profile of high-affinity but not low-affinity receptors.

### High-affinity CAR-T cells show fewer signs of toxicities and exhaustion when combined with low-affinity CARs in vitro

Next, we explored the possibility of creating a CAR-T-cell product with improved characteristics by combining the potency of high-affinity receptors with the safety profile and longevity of lower-affinity receptors. For this purpose, we evaluated the cytokine release of affinity-combined CAR-T-cell products composed of an equal amount of the high-affinity JCAR017-T cells and lower-affinity CAR-T cells (JCAR021, S168C or S227K) after co-culture with CD19+ target cells and donor-matched monocytes. These mixed products were compared with two JCAR017-T-cell controls: one matched for the total CAR-T-cell dose and one matched for the number of JCAR017-T cells present in the affinity-combined product. (Fig. 5a). To distinguish between CAR constructs, an Ametrine fluorescent protein was co-expressed with the JCAR017-CAR. After 24 h of co-culture, absolute cell numbers of CAR-T cells were comparable among the different groups (Fig. 5b), with a balanced composition of the high- and low-affinity CAR-T cells in the combined products (Fig. 5c). While all CAR groups showed similar CD19-specific target lysis, the mixed products induced significantly lower levels of CRS-associated cytokines than the same total dose of JCAR017-CAR T cells alone and exhibited cytokine release comparable to the half-dose JCAR017 control (Fig. 5d, e). These observations indicate that adding lower-affinity CAR-T cells does not increase

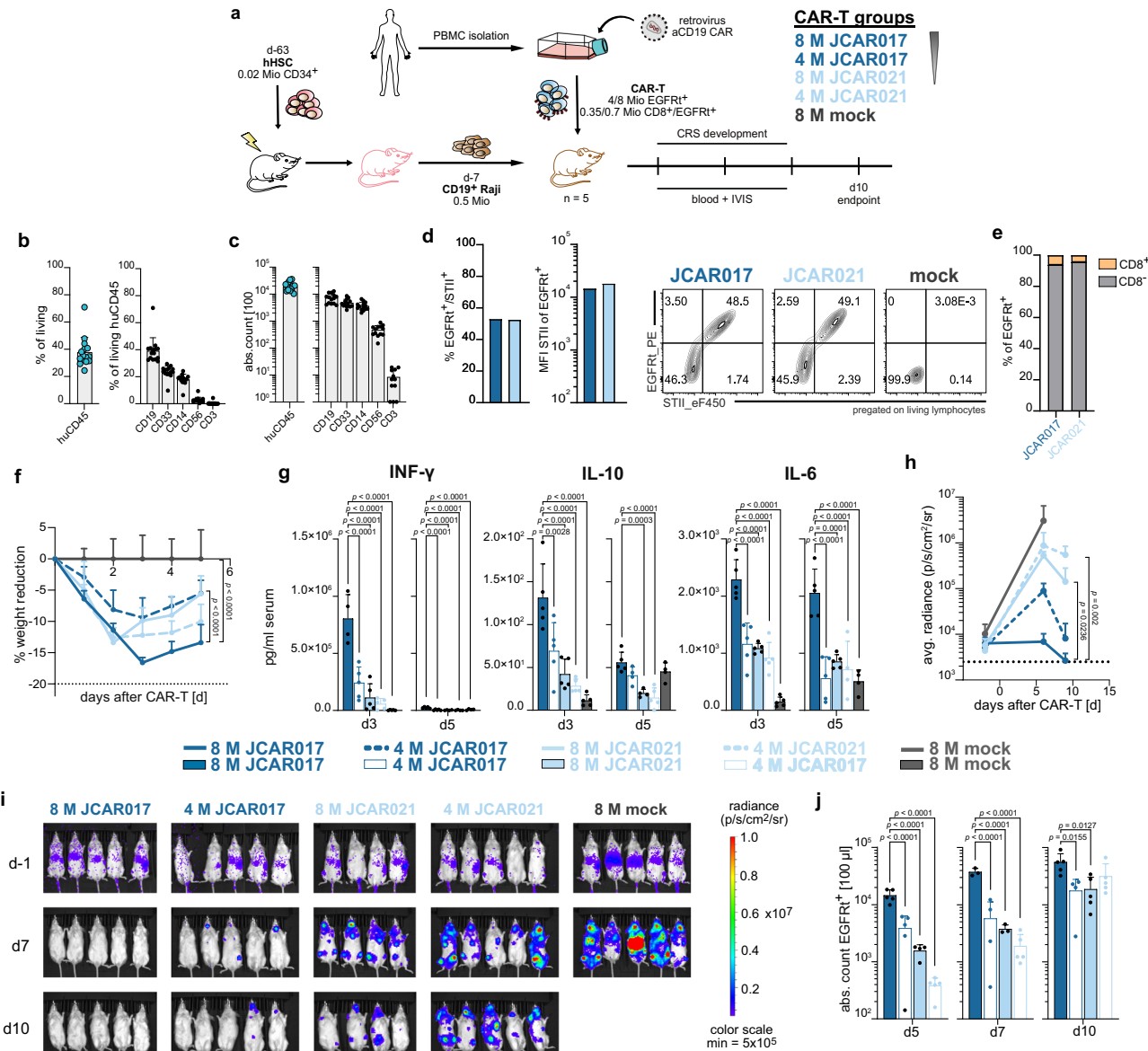

**Fig. 4 | CAR-T-cell-associated toxicities are influenced by T-cell dose and receptor affinity. a** Schematic representation of the experimental design for the humanized CRS mouse model employing sorted CAR-T cells. The overall design is consistent across all CRS experiments, while the specific CAR-T-cell transfer conditions illustrate comparisons between varying doses of high- and low-affinity CAR-T cells. **b** Frequencies of huCD45+ chimerism and immune subpopulations in peripheral blood. **c** Absolute cell counts of huCD45+ cells and immune subpopulations in 100 µL of blood. **d** Transduction efficiencies of JCAR017 and JCAR021 constructs, shown as frequencies of EGFRt+/STII+ cells (left), STII mean fluorescent intensity (MFI) within EGFRt+ T cells (middle) and representative flow cytometry plots (right)

of the infusion products. **e** Distribution of CD8+ to CD8− population within CAR-transduced cells in the infusion product. **f** Weight reduction following a high- and low-dose of high-affinity JCAR017 and low-affinity JCAR021-T cells. **g** Serum cytokine levels (IFN-γ, IL-10 and IL-6) three and five days after CAR-T-cell transfer. **h–i** Quantification (**h**) and imaging (**i**) of residual tumor burden of Raji cells determined via bioluminescence imaging. **k** CAR-T-cell expansion in blood (EGFRt+ cells per 100 µl) on day 5, 7, and 10 post-transfer. Each dot depicts an individual mouse. Data represent mean + SD (n = 5). Statistical analysis was performed by two-way ANOVA (**f, h**) or by one-way ANOVA (**g, j**), using the JCAR017 group with the highest dose as a reference control. *p < 0.05, **p < 0.01, ***p ≤ 0.001, ****p ≤ 0.0001.

toxicity beyond what is expected from reducing high-affinity CAR T-cell numbers.

We then aimed to investigate the potential benefits of an affinity-combined CAR-T-cell product after prolonged antigen exposure. To do this, we performed chronic antigen stimulation assays and assessed expansion and longevity potential of the individual CARs compared to the affinity-combined T-cell products (Fig. 5f). The combined-affinity products expanded similarly to high- and lower-affinity CAR-T cells alone (Fig. 5g). Notably, lower-affinity CAR-T cells showed higher expansion kinetics under chronic stimulation regardless of the supplementation of high-affinity CAR-T cells, thus reaching the majority in

the combined cell product (Fig. 5h, i). We then examined the expression of PD-1, Lag3, TIM3, and TIGIT to assess the level of exhaustion. All four markers showed significantly higher levels in the high-affinity JCAR017-T cells compared to both the lower-affinity and combined T-cell products (Fig. 5j). Remarkably, the proportion of TIGIT+ TIM3+ and TIGIT+ PD-1+ cells was reduced in JCAR017-T cells when used in combination with lower-affinity T cells, as opposed to when JCAR017-T cells were used individually (Fig. 5k). Finally, we assessed whether the affinity-combined product can retain similar anti-tumor activity to high-affinity CAR-T cells alone. We therefore focused on the two ultra-low-affinity variants (V236W and V236F) that exhibited impaired

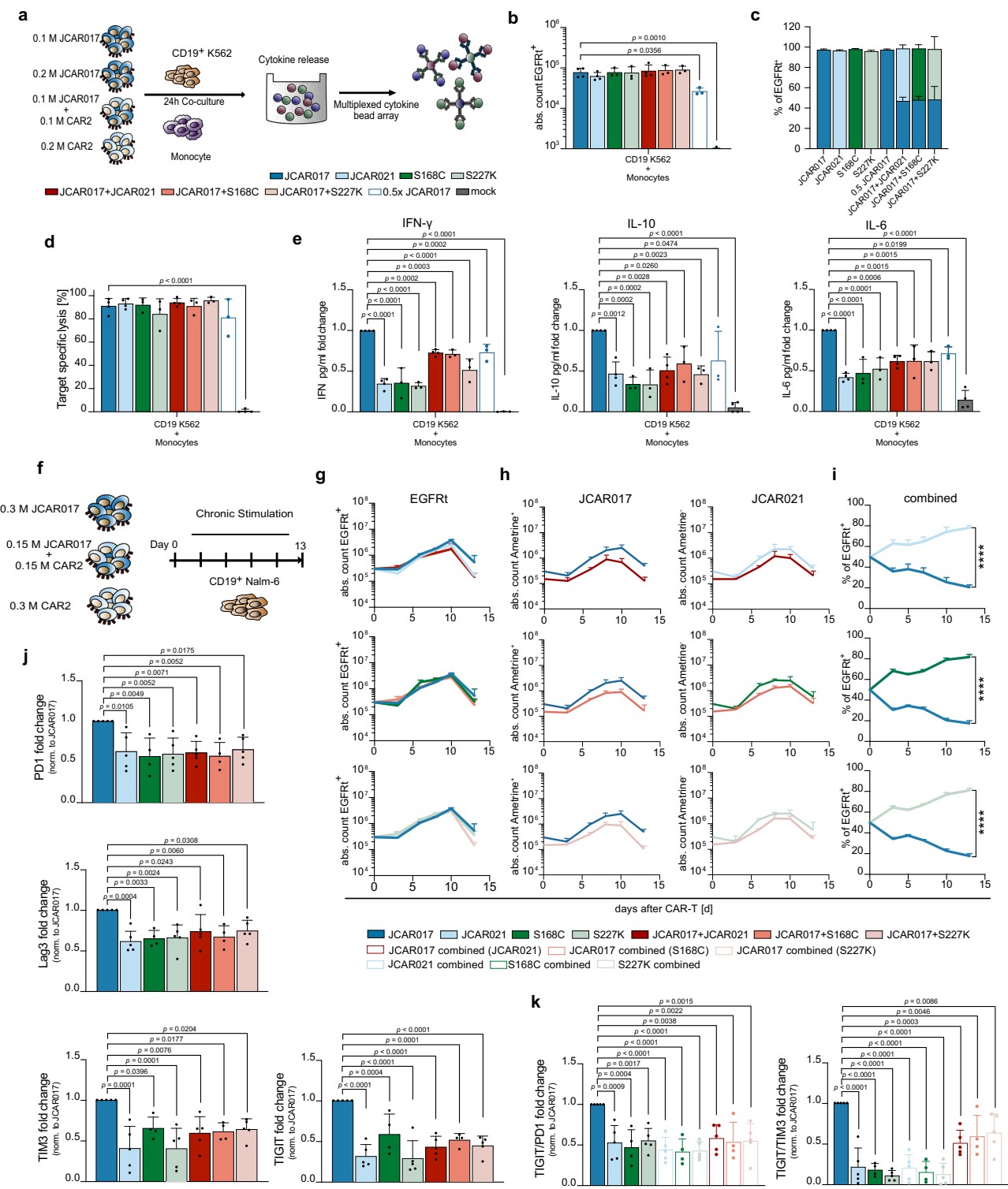

in vitro cytotoxicity under conditions of low E:T ratios and low antigen density (Fig. S4). Notably, the combined products rescued the suboptimal cytotoxic capacity observed with low-affinity CAR-T cells alone, achieving levels of target cell killing comparable to the same dose of high-affinity CAR-T cells but exceeding those of the 0.5× dose JCAR017 control, particularly under stringent conditions of low E:T ratios and low antigen density (Fig. S10). To extend our findings, we evaluated an additional pair of high- and low-affinity CARs recognizing the same epitope on the human ROR1 antigen—CAR R12 (high-affinity, KD = 0.11 nM) and CAR R12v20 (lower-affinity, KD = 0.98 nM)[57] (Fig.

S11a). Similar to the CD19 model, a 50:50 combination of high- and low-affinity CAR-T cells achieved cytotoxicity comparable to full-dose high-affinity R12 CAR-T cells while outperforming lower-affinity R12v20 CAR-T cells (Fig. S11b, c), and exhibited significantly lower exhaustion levels than high-affinity CAR-T cells after chronic stimulation (Fig. S11d).

Overall, the affinity-combined cell product mitigated CRS severity, limited exhaustion in high-affinity CAR-T cells, and preserved functionality, with particular resilience against functionality loss when targeting cells with low antigen expression.

**Fig. 5 | High-affinity CAR-T cells show less signs of toxicities and exhaustion when combined with low-affinity CAR-T cells in vitro. a–e** CAR-T cells were co-cultured with CD19$^+$ K562 and donor-matched monocytes using a ratio of 2:1:1 for 24 h prior analysis. **a** Schematic of the in vitro experimental setup for evaluating CRS. **b** Absolute CAR-T-cell counts and **c** frequencies of Ametrine$^{+/-}$ across treatment groups. **d** CD19$^+$ K562 target cell lysis. **e** Supernatant cytokine concentrations (IFN-γ, IL-10 and IL-6). Data are background-subtracted normalized to JCAR017. **f–k** CAR-T cells were cultured with CD19$^+$ Nalm-6 cells for 13 days, with repetitive supplementation of fresh target cells every second day. **f** Schematic of the experimental setup for chronic antigen exposure. **g** Expansion kinetics of differently composed CAR-T-cell products. **h** Expansion of high-affinity (JCAR017) and lower-affinity (JCAR021, S168C, S227K) CAR-T cells cultured individually or in combination based on the expression of Ametrine. **i** Temporal composition of the affinity-combined CAR-T-cell products. Data in (**g–i**) are depicted as mean + SD of one representative experiment with technical triplicates (*n* = 4–5). **j** Expression of exhaustion markers (PD1, Lag3, TIM3 and TIGIT) following chronic antigen exposure at 13 days post-exposure, following five antigen addition cycles. **k** Co-expression of TIGIT/PD1 and TIGIT/TIM3 on JCAR017 and JCAR021 CAR-T cells alone or in combination. Data in (**b–e**) are represented as mean + SD of (JCAR017 and JCAR021, combination *n* = 4, S168C, S227K, combination and 0.5 JCAR017 *n* = 3) biological replicates. Data in (**e**) and (**j, k**) are normalized to JCAR017 and shown as mean + SD. Each individual dot represents the mean of three technical replicates (*n* = 5; S168C *n* = 4). Statistical analyses were performed by one-way ANOVA (**b, d, e, j, k**), or two-way ANOVA (**g–i**) with multiple comparison using JCAR017 as reference. *\*p* < 0.05, *\*\*p* ≤ 0.01, *\*\*\*p* ≤ 0.001, *\*\*\*\*p* ≤ 0.0001 using the JCAR017 group as a reference control.

## High-affinity CAR-T cells show superior functionality and fewer signs of exhaustion when administered with low-affinity CARs in vivo

We next explored the potential of an affinity-combined CAR-T-cell product in vivo. We first evaluated whether the suboptimal tumor control observed with low-dose high-affinity CAR-T cells could be rescued by higher doses of low-affinity CAR-T cells (Fig. S12a). This strategy was designed to minimize CRS, as CRS severity correlated with the dose of high-affinity—but not low-affinity—CAR-T cells (Fig. 4). While JCAR017-T cells alone delayed tumor growth, the combination with JCAR021-T cells produced a dose-dependent reduction in tumor burden. Nevertheless, a significant effect emerged only at particularly high doses of JCAR021-T cells (16-fold higher than JCAR017) (Fig. S12b, c), indicating the importance of maintaining adequate levels of high-affinity CAR-T cells for achieving effective anti-tumor responses.

Furthermore, we conducted experiments applying constant cell doses, consisting of either equal amounts or varying ratios of high- and low-affinity CARs (Figs. 6a and S13a). As observed in our previous experiments, survival did depend on the transfer of even low doses of JCAR017-T cells (Fig. 6b). Residual tumor burden further decreased based on the quantity of transferred JCAR017-T cells (Figs. 6c and S13b). Remarkably, a 1:1 ratio of JCAR017 to JCAR021 exhibited a more rapid anti-tumor response compared to JCAR017 alone (Figs. 6c and S13b), which we validated in an independent experiment (Fig. S12d, e). While JCAR017-T cells exhibited dose-dependent expansion kinetics, the same dose ($0.8 \times 10^6$ cells) expanded differently depending on whether it was administered alone or in combination with low-affinity CAR-T cells, with JCAR017-T cells expanding and infiltrating the bone marrow less when combined with JCAR021-T cells (Figs. 6d and S13c). These findings indicate that the observed faster tumor clearance mediated by the affinity-combined product is likely attributable to a synergistic interaction between the high- and low-affinity CAR-T cells, rather than by the overall higher T-cell dose. Our data also demonstrate that the behavior of high-affinity CAR-T cells can be modulated by addition of lower-affinity CAR-T cells, even when the high-affinity CAR T-cell dose remains at high numbers.

To explore this further, we isolated the transferred CAR-T cells 30 days after infusion to ensure sufficient antigen exposure to capture relevant differences in the transcriptomic profile of JCAR017-T cells. Remarkably, the transcriptomic profiles of JCAR017-T cells strongly differed depending on whether the cells were administered individually or in combination with low-affinity CARs. CAR-T cells isolated from mice treated with JCAR017 alone were predominantly represented in cluster 1, whereas high-affinity JCAR017-T cells of the combined cell products were in cluster 0 and 6 (Figs. 6e and S13d, e). Genes overexpressed in JCAR017-T cells within the affinity-combined products collectively suggested active migration toward inflammatory sites and tissue residency (*CXCR6* and *ITGA1*)[58], together with sustained pro-inflammatory/cytotoxic responses (*ALOX5AP, KLRD1, STAT, and JAK3*). Along these lines, we further observed upregulation of genes related to increased energy demand and activation (*MT-CO3* and *FKBP1*)[59,60]. In contrast, JCAR017-T cells alone exhibited traits of impaired memory T-cell formation and survival (downregulation of *CD27*, *KLF2*, and *TNFSF4*), metabolic shifts towards glycolysis (lower *CMC1*)[61] and an enhanced effector phenotype (reduced *IL10RA*, and *SIRPG*)[62–64]. Altogether, this profile indicated highly activated effector T cells prone to an immediate response rather than sustaining long-lived memory cells (Fig. 6f). Notably, when considering a set of 262 genes linked to T-cell exhaustion and dysfunction[65,66], JCAR017-T cells administered alone showed higher levels of several exhaustion-related genes compared with the affinity-combined cohorts, which displayed upregulation of only a few exhaustion-associated genes and showed a profile more similar to the lower-affinity CAR-T cells (Fig. 6g). Interestingly, the shift toward a less exhausted phenotype in high-affinity JCAR017-T cells became progressively more pronounced from the 10:90 to the 50:50 combination ratios. This trend is likely attributable to increased antigen pressure on JCAR017-T cells when delivered at lower doses, as further supported by the higher residual tumor burden observed under these conditions (Fig. 6c). Accordingly, key exhaustion-associated genes, such as *CCL5* and *TOX*, were expressed at significantly lower levels in JCAR017-T cells from the 50:50 combination (Fig. 6h). Conversely, the expression of genes implying greater cytotoxic potential (*KLRD1*, *SCML4* and *GZMB*) was increased in affinity-combined JCAR017-T cells (Fig. 6i), resulting in reduced exhaustion and dysfunctional scores, most notably in the 50:50 dosing condition (Fig. 6j). These data were in line with in vitro observations (Fig. 5j, k).

Overall, our findings suggest that the enhanced tumor clearance observed with the affinity-combined product could be driven by a synergistic interaction between high- and low-affinity CAR-T cells, rather than by the overall higher T-cell dose. Our data also demonstrate that the behavior of high-affinity CAR-T cells can be modulated by the addition of lower-affinity CAR-T cells, even when the high-affinity CAR-T-cell dose remains at high numbers. Collectively, our data demonstrate that low-affinity CAR-T cells augment the anti-tumor efficacy of high-affinity CAR-T cells by preserving their functional state and mitigating terminal differentiation and exhaustion.

## Discussion

In cancer immunotherapy, the use of engineered T cells expressing CARs has proven highly effective in treating B-cell-mediated diseases[67,68]. However, challenges such as safety and impaired long-term functionality remain relevant. These issues can be partially attributed to the use of high-affinity receptors, which trigger supraphysiological activation of CAR-T cells favoring exhaustion[69], enhancing the selection towards antigen loss and fratricide[28], and contributing to more severe acute toxicities[25]. While lower-affinity CARs seem to mitigate these phenotypes[25,70], identifying the optimal affinity range to balance functionality and safety remains unknown.

In this study, we first systematically analyzed the impact of CAR binding affinities on CAR-T-cell functionality and safety. We employed two CAR constructs targeting the same epitope on the CD19 antigen: JCAR017 (derived from the FMC63 antibody clone), a high-affinity

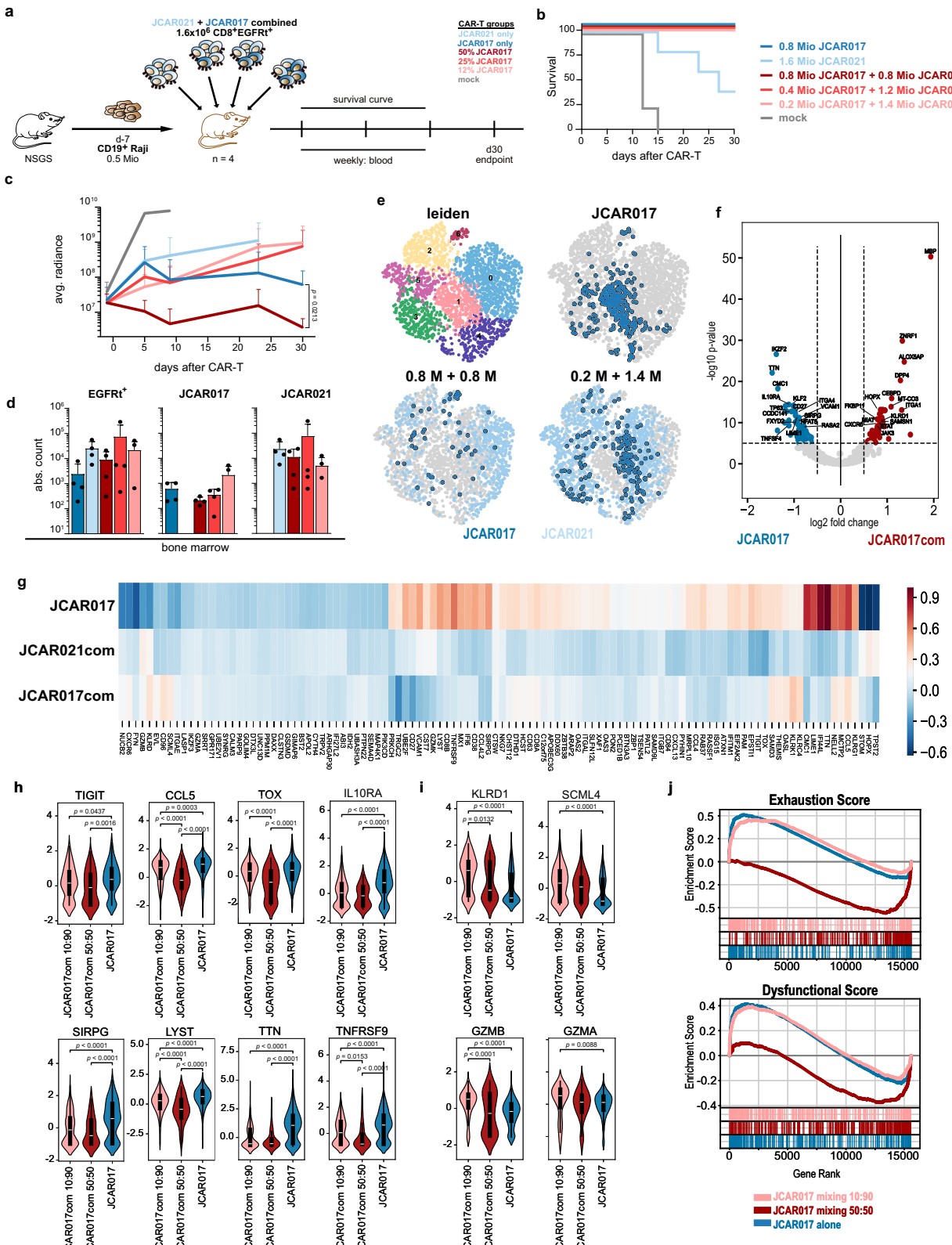

**Fig. 6 | High-affinity CAR-T cells show superior functionality and less signs of exhaustion when administered with low-affinity CARs. . a** Experimental design for in vivo evaluation of affinity-combined CAR-T-cell products employing bulk CAR-T cells (*n* = 4). **b** Kaplan-Meier survival curve of mice treated with varying JCAR017 and JCAR021 combinations. **c** Residual tumor burden assessed by bioluminescence imaging (photons/second/cm²/steradian), shown as mean + SD.
**d** Absolute CAR-T-cell numbers, including JCAR017 and JCAR021 isolated from the bone marrow (BM) at sacrifice. **e** Leiden clustering of EGFRt⁺ CAR-T cells isolated from BM of mice treated with JCAR017 alone or affinity combinations (50:50 and

90:10). **f** Volcano plot depicting upregulated genes in JCAR017 (blue) and JCAR017 form combined products (red). **g** Heatmap of exhaustion-related genes (*n* = 110) in JCAR017, as well as JCAR021 and JCAR017 from combined products. **h** Differential expression of exhaustion-related and **i** functionality-related genes in JCAR017 alone and JCAR017 of combined products. **j** Gene set enrichment analysis depicting exhaustion (top) and dysfunction (bottom) scores based on literature-defined gene sets. Each dot depicts an individual mouse. Data represent mean + SD (*n* = 4). Statistical significance was determined by two-way ANOVA (c) with JCAR017 as reference and Mann-Whitney U test (h-i). *p < 0.05, **p ≤ 0.01, ****p ≤ 0.0001.

binder and JCAR021, a lower-affinity receptor. Additionally, we generated a library of JCAR021-derived mutants covering an unprecedented spectrum of affinities, even as low as binding affinities of TCR:pMHC interactions, while maintaining effective CAR-T-cell activation.

We observed that the majority of CAR constructs (JCAR017, JCAR021 and CAR[mfunct]) preserved in vitro CAR-T-cell functionality, despite exhibiting approximately a 40-fold variation in $t_{1/2}$, in line with previous findings[71]. Among this group, only two mutants with TCR-like binding kinetics, V236W and V236F, exhibited a modest reduction in cytokine production and impaired cytotoxicity, particularly against targets with low antigen density. Furthermore, we identified a set of mutants (termed CAR[mlowfunct]) with moderate to low functionality in vitro, spanning various values across the affinity spectrum. However, these mutants showed clear signs of reduced/impaired CAR surface expression (unstable or undetectable Strep-Tag signal), which likely represents a major cause for the observed functional impairment, as shown by others[72]. It is known that single-point mutations can alter protein folding, protein stability or receptor processing to the cell surface[73]. Therefore, we recommend determining receptor surface expression in screening protocols searching for receptor modifications which aim to alter antigen-binding characteristics.

In contrast to in vitro assays, in vivo experiments segregated a precise correlation between CAR-T-cell performance and antigen-binding affinity. High-affinity binders demonstrated the highest efficacy, while JCAR021-T cells could only delay tumor growth, presumably due to reduced engagement in the anti-tumor response, as evidenced by lower bone marrow infiltration and differentiation. While lower binding affinities have been shown to enhance CAR-T-cell functionality[25,42,43], this enhancement depends on maintaining a binding threshold sufficient for proper CAR functionality[72], which appears not to have been met by our JCAR021 CAR clone. Ultra-weak, TCR-like binders turned out to be almost ineffective in controlling tumor progression, confirming the lower sensitivity of CAR-T cells compared to TCR-engineered or physiological T cells[74,75].

Currently, only highly sophisticated and technically demanding humanized mouse models are capable to analyze CAR-T-cell toxicities like CRS or ICANS. Here, we further developed the NSG-S model to generate larger cohorts of humanized mice rederived form single human HSC cord blood donors, in order to provide more homogenous experimental conditions[56]. With the help of these preclinical models, we could very clearly elaborate that high-affinity CAR-T cells are more prone to mediate severe CRS toxicities than CAR-T cells with lower antigen-binding affinities. Moreover, CRS mediated by high-affinity CAR-T cells is dose-dependent, very similar to published clinical dose-escalation studies[11], where the administration of reduced CAR-T-cell doses also correlated with a decreased risk of severe toxicities[11]. However, given the significant variability in tumor burden and antigen load between individual patients, determining the optimal dosage of high-affinity CAR-T cells prior to therapy presents a considerable challenge, and may be difficult to achieve with precision as of now.

Reduction of CAR antigen affinity has been suggested before as a strategy to generate a broader therapeutic window with the maintenance of high anti-tumor efficacy while reducing the risk of adverse events and CRS occurrence even when relatively high numbers of engineered CAR-T cells are adoptively transferred into patients. An intriguing finding of our study is that low-affinity CAR T cells, very differently from high-affinity CAR-T cells, can be applied at much larger doses without necessarily increasing side effects, but remarkably displayed dose-dependent improvements in anti-tumor cytotoxicity. This is in line with observations derived from the first clinical testing of low-affinity CARs involving low tumor burdens[25]. However, despite the improved safety profile and significant anti-tumor activity, low-affinity CARs such as JCAR021, mediated impaired anti-tumor clearance. Other groups have similarly reported reduced functionality with decreasing receptor affinity[57]. These findings highlight the advantages but also

limitations of pure high- and low-affinity CAR-T products in clinical applications.

Similar to the core principles underlying protective polyclonal T-cell responses, which is characterized by the activation and expansion of a diverse repertoire of T-cell clones of high- and low-affinity TCRs, we proposed that creating an affinity-combined CAR-T-cell product might offer a promising approach to ensure high safety without compromising—or potentially even harming—efficacy. Therefore, we combined two very different antigen-binding affinities (JCAR017 and JCAR021) within a single CAR-T-cell product, aiming to leverage the immediate efficacy of high-affinity CARs, while integrating the safety and activation benefits of low-affinity CARs.

First in vitro experiments strongly supported the concept of affinity combination, as this setting reduced the production of monocyte-derived IL-6, a critical cytokine that mediates monocyte bystander effects and the severity of CRS at increasing doses[18]. Furthermore, the production of inflammatory cytokines like IFN-γ and IL-10 was reduced for CAR-T-cell products with combined-affinity. In chronic stimulation assays, prolonged antigen exposure skewed the composition of the affinity-combined CAR-T-cell product, with low-affinity CAR-T cells dominating over time—a phenomenon akin to natural immune responses[26,76]. This effect is likely attributed to the propensity of high-affinity CAR-T cells to be more susceptible to exhaustion and activation-induced cell death[71]. In contrast, the extended persistence of lower-affinity CAR or TCR binders is likely attributed to the propagation of better persisting weakly differentiated memory T-cell subsets[25,26]. Moreover, the increased extent of trogocytosis and the resulting fratricide-induced cell death observed in high-affinity CARs may contribute to the progressive dominance of lower-affinity CARs over time[28].

Affinity combination in tumor-bearing NSG-S mice recapitulated our in vitro observations. High-affinity CAR-T cells exhibited superior anti-tumor efficacy compared to low-affinity CAR-T cells, confirming that a minimum amount of the former remains fundamental for sustained tumor responses. Furthermore, combined-affinity CAR-T-cell products demonstrated a significantly stronger effect in tumor reduction. We observed that such an effect is not attributable to cell doses, as we found even lower initial expansion of high-affinity JCAR017-T cells in the combined product compared to high-affinity CAR-T cells alone, consistent with our in vitro findings. Single-cell transcriptomic analysis revealed that JCAR017-T cells showed traits of sustained pro-inflammatory/cytotoxic responses along with lower levels of exhaustion-associated genes in the combined product compared to monoclonal transfer, further supporting the beneficial impact of the affinity combination strategy. As the two CARs recognize the same epitope, we might speculate that the low-affinity CAR-T cells could limit antigen availability, thereby restraining the engagement of high-affinity CAR-T cells and thus tuning their activation. However, further experiments will be required to elucidate the underlying mechanism. Given that early CAR-T-cell expansion correlates with the severity of adverse events[12,77], the affinity-combined product should be associated with improved treatment safety. Noteworthy, it was described that high-affinity anti-CD19 CAR-T cells contribute significantly to antigen loss across several tumor types, either approved for or under investigation in CAR-T-cell therapy[28]. This raises concerns about the potential persistence of antigen-escape variants. Both antigen-escape and trogocytosis appear to be affinity-dependent[28,78].

Given the challenges in identifying low-affinity constructs that maintain efficacy and the considerable intra-tumoral heterogeneity in antigen expression observed in human cancers, a combined CAR-T-cell product may offer the best balance between sensitivity to low antigen densities and reducing selective pressure on tumor cells. We systematically elaborated many features of CARs with different affinities. Thereby, we found, that especially lower-affinity CAR-T cells still exhibit strong on-target functionality but are less activated,

differentiated and effective at clearing tumors, in particular with low antigen expression. However, in affinity-combined CAR-T-cell products, lower-affinity CAR-T cells mitigate excessive activation, expansion and exhaustion of high-affinity CAR-T cells, thereby contributing to improved therapeutic outcomes.

In conclusion, our findings demonstrate the presence of a minimum affinity threshold required for CAR-T cells to function effectively for currently approved engineering formats, which is significantly higher than that for functional TCRs. Beyond this threshold, efficacy correlates with antigen-binding strength in our model, though this comes at the cost of compromising safety. An innovative affinity-combination approach offers a promising strategy to mitigate the limitations of affinity-reduced CAR-T cells while addressing the risks associated with high-affinity CAR-T cells, including CRS severity, rapid exhaustion, and antigen escape. Our study shows that combining CAR-T cells with varying affinities for the same CD19 antigen reduces exhaustion and toxicity, probably via synergistic effects, thereby promoting proliferative capacity and enhancing anti-tumor responses, particularly in conditions of chronic antigen exposure. This beneficial effect is likely of multifactorial mechanistic origin, with a potential primary contributor being the competitive interaction of lower-affinity CAR-T cells for the same antigen, ultimately alleviating activation-induced stress on high-affinity counterparts. Our data shows that high-affinity CAR-T cells remain functionally active in the presence of lower-affinity CAR-T cells, leading to a more controlled and sustained immune response. This approach has the potential to expand the therapeutic window of CAR-T-cell therapies and to improve their efficacy against malignancies expressing tissue-specific antigens.

## Limitations of the study

This study provides valuable insights into potential advancements in CAR-T-cell therapy by determining the optimal ligand binding affinity and affinity combinations. However, some limitations should be noted to contextualize the findings and guide future research. The concepts explored in this study are currently applicable mainly to anti-CD19 CAR constructs incorporating a 4-1BB co-stimulatory domain, despite additional proof-of-principle data with anti-ROR1 CARs indicating a more general phenomenon. To extend these findings to other therapeutic options, further investigations are required with alternative binders and co-stimulatory domains, particularly given the distinct characteristics of CD28-based CARs (e.g., enhanced activation and reduced longevity). Another limitation of our study is that we have not determined the exact lower affinity threshold for CAR-T cells where the beneficial effect of affinity-combined T-cell products disappears. This is experimentally highly challenging and will require future studies. Moreover, further experiments are required to elucidate the mechanisms underlying the beneficial effects of affinity-combined CAR-T-cell products, including analyses of high-/low-affinity CAR pairs targeting distinct epitopes of the same antigen. Finally, the use of different experimental models—immunocompromised tumor-bearing mice for long-term studies versus humanized mice for CRS modeling, as well as Nalm6 versus Raji tumor cells—necessitated the use of different CAR-T-cell doses across in vivo experiments, thereby limiting definitive conclusions regarding the optimal ratio of the affinity-combined product. Our data nevertheless strongly suggest that an equal ratio of high- and low-affinity CAR-T cells has the potential to balance functionality, safety, and persistence. However, additional studies will be required to further substantiate this concept.

## Methods

All experiments were conducted in accordance with all relevant ethical regulations. The study protocol involving human samples was approved by the local Institutional Review Board (Ethikkommission der Medizinischen Fakultät der Technischen Universität München). The conducted animal experiments were approved by the local authorities, especially the district government of Upper Bavaria (Department 5—Environment, Health and Consumer Protection ROB-55.2-2532.Vet_02-18-162).

## Cell culture

Cell lines (Raji-ffluc-GFP, Nur77-Jurkat kindly provided by Juno Therapeutics GmbH, a Bristol Myers Squibb company; Nalm6-ffluc-GFP, RD114, HEK293T cells kindly provided by the Stanley Riddell laboratory at the Fred Hutchinson Cancer Research Center, Jeko-1 ffluc and A549 ffluc cells kindly provided by the Michael Hudecek laboratory at the Department of Internal Medicine II, University Hospital Würzburg) were cultivated in Gibco RPMI 1640 medium (Thermo Fisher Scientific) or Gibco DMEM (Thermo Fisher Scientific) at 37 °C, 5% $CO_2$, and 95% humidity. Both media were supplemented with 10% fetal bovine serum (FBS; Sigma Aldrich), 0.025% L-glutamine (Gibco BRL), 0.1% HEPES (Gibco BRL), 0.001% gentamycin (Gibco BRL) and 0.002% streptomycin (Gibco BRL) ("complete medium", cRPMI or cDMEM).

Primary T cells were derived from healthy donors. Written informed consent was obtained from the participants, and usage of the blood samples was approved according to national law by the local Institutional Review Board (Ethikkommission der Medizinischen Fakultät der Technischen Universität München). Blood was diluted 1:1 (whole blood) or 1:2 (buffy coat) with sterile PBS and peripheral blood mononuclear cells (PBMC) were isolated by density gradient centrifugation using Pancoll solution (density 1.077 g/mL) (PanBiotech) according to manufacturer's protocol. For engineering, human T cells were cultured at a density of $1 \times 10^6$ cells/mL in complete RPMI and activated with 2.25 µl/mL anti-CD3/CD28 Expamer Reagent (Juno Therapeutics GmbH, a Bristol Myers Squibb company)[79] as well as 360 IU/mL IL-2 (Peprotech) for 48 h. Jurkat cells were thawed and cultured for one week. For in vitro cell culture, bulk T cells were cultured at a density of $1 \times 10^6$ cells/mL in complete medium supplemented with 50 IU/mL IL-2. For rapid expansion, T cells were cultured at a density between $0.25-0.5 \times 10^6$ cells/mL in complete RPMI supplemented with 200 IU/mL IL-2, 0.5 ng/mL IL-7 and 0.5 ng/mL IL-15.

Human HSC were derived from umbilical cord blood. After cord blood mononuclear cells (CBMC) isolation via density gradient centrifugation using Pancoll solution (density 1.077 g/mL), CD34+ HSCs were enriched by magnetic cell isolation using the CD34 Microbead Kit (Miltenyi Biotech) according to the manufacturer's protocol. All cells were stored in freezing medium (10% DMSO and 90% FBS; Sigma Aldrich) for cryopreservation at a maximum concentration of $10 \times 10^6$ cells/mL.

## DNA template design for scFv periplasmic protein production and CAR transduction

scFv sequences of JCAR017 (clone: FMC63) and JCAR021 were kindly provided by Juno Therapeutics GmbH, a Bristol Myers Squibb company. scFv sequences of the clone CAT was obtained from the publicly available patent (WO 2014/184143A1). scFvs were generated by fusing the variable regions of the heavy (VH) and light (VL) chains with a short (G4S)3 linker. DNA templates were designed in silico and synthesized by GeneArt (Thermo Fisher Scientific) or Twist Bioscience. Anti-ROR1 CARs (R12 and CDR3 mutated R12v20) were kindly provided by the Michael Hudecek laboratory at the Department of Internal Medicine II, University Hospital Würzburg. Both ROR1 were constructed using the VL and VH segments of the R12. The scFv was linked by a (G4S)3 linker, a CD28 transmembrane domain and a 4-1BB cytoplasmic domain.

For periplasmic expression, scFv sequences were linked to a Twin-Strep-tag and a Tub-tag sequence building so-called 'scFv FLEXamers'[54]. For CAR constructs, the signal peptide of GM-CSF receptor subunit α, which enables the transport and integration of the receptor into the membrane, was followed by an anti-CD19 scFv. The extracellular binding domain was linked to a spacer domain comprising a triple repetitive sequence of Strep-tag II (STII)[46,48] and parts of the

IgG4-Fc molecule, followed by a transmembrane region originated from the CD28 chain. The following intracellular signaling domains, CD3-zeta and 4-1BB, were separated by a viral T2A peptide from a truncated version of EGFR (EGFRt) used as a marker of template integration[45,47]. To distinguish JCAR017 and JCAR021 in affinity combination experiments, the sequence of the fluorescent protein Ametrine and mCherry were included on the C′-terminus and connected by a T2A element.

## Generation of JCAR21 scFv mutants

The three-dimensional structure of the extracellular JCAR021 scFv binding domain was modeled in silico with the ABodyBuilder software by SAbPred[80], and applied to the webserver mCSM-AB[81] for the prediction of mutational affinity changes. The delta in Gibbs free energy ($\Delta\Delta G$) of JCAR021 after binding to the extracellular domain of the CD19 protein was calculated for all possible single amino acid substitutions within the JCAR021 scFv. Out of 231, 32 were selected for further analysis of reduced receptor binding affinity, with 19 substitutions being located in the framework region and 14 in the CDRs region (Tab. S1). While amino acid exchanges within the framework region were chosen based on the alteration in Gibbs free energy, those in the CDRs were only considered when similar biochemical properties (groups: hydrophobic, polar, acidic and nonpolar) were retained to preserve CAR specificity (Tab. S1). The selected amino acid substitutions were integrated via Q5 site-directed mutagenesis PCR (New England Biolabs GmbH) and subsequently cloned into the pMP72 and pASG-IBAwt2 vectors for retroviral transduction and periplasmic protein production, respectively. The pMP72 vector was derived in-house from the pMP71 vector (a kind gift from W. Uckert, Max-Delbrück-Centrum for Molecular Medicine, Berlin).

## Recombinant protein expression of soluble scFvs

scFv DNA strings were cloned into a pASG-IBAwt2 vector (IBA Lifesciences) and expressed in electrocompetent *E. coli JM83* (IBA Lifesciences). Expression was induced with anhydrotetracycline (IBA Lifesciences) (AHT−1:10^5; IBA Lifesciences) at an $OD_{600}$ of 300 for at least 3 h or as soon as $OD_{600}$ values stabilized at 37 °C. Bacteria were centrifuged (5000 rpm, 4 °C, 12 min), and the pellets were stored at −80 °C. Lysis was performed in 10 mL periplasmic lysis buffer (100 mM Tris/HCl pH 8.0−Roth, 500 mM Sucrose−Sigma Aldrich, 1 mM EDTA−Sigma Aldrich) (30 min, 4 °C) followed by debris centrifugation (15,000 rpm, 4 °C, 15 min), nucleic acid digestion in 125 U Benzonase (Sigma Aldrich) (1 h, 4 °C, rolling) and scFv purification on *Strep*-Tactin superflow columns (IBA Lifesciences) according to manufacturer's instructions. Protein concentrations were measured via spectrophotometry at 280 nm, and purity was determined via SDS-PAGE. Purified scFv protein aliquots were shock-frozen in liquid nitrogen and stored at −80 °C.

## Retroviral transduction

CAR DNA templates were cloned into the retroviral expression vector pMP72. The mp72 vector was generated in-house by modifying the multiple cloning site of the mp71 vector to introduce specific restriction sites. The mp71 vector was kindly provided by W. Uckert (Max-Delbrück-Centrum für Molekulare Medizin, Berlin). The RD114 packaging cell line was seeded at a density of $1 \times 10^6$ cells in 3 mL of cDMEM per well of a six-well plate and allowed to adhere for 18 h. At approximately 80 % confluency, the cells were transiently transfected via calcium phosphate precipitation using 18 µg DNA template followed by a medium exchange after 6 h of incubation. After 3 days, virus-containing supernatant was harvested, filtered through a 0.45 µm filter and used directly or stored at 4 °C for up to one month. Tissue-culture untreated 24-well plates were coated overnight with 300 µl RetroNectin (Takara Bio Europe, 1:100 in sterile PBS) according to the manufacturer's protocol. 500-700 µl of virus-containing

supernatant was added and centrifuged at 2000 g at 32 °C for 2 h, followed by the addition of $0.5 \times 10^5$ activated cells (see 'Cell culture" section) in 200 µl RPMI medium. Primary human T cells were supplemented with 200 IU/mL IL-2. Plates were centrifuged at 800 g, 32 °C for 30 min and incubated at 37 °C for 48 h before further analyses. Primary human CAR-T cells were expanded as described earlier and used no later than three weeks after generation for functional in vitro and in vivo characterization.

## Lentiviral transduction

Lentiviral particles were produced in HEK293T cells using transient transfection. A total of $1 \times 106$ cells were seeded one day prior to transfection to reach approximately 70-80 % confluency. Cells were co-transfected with the lentiviral transfer plasmids together with the packaging plasmids psPAX2 (gifted from Didier Trono; Addgene plasmid #12260) and pMD2.G (gifted from Didier Trono; Addgene plasmid #12259) using Lipofectamine (Thermo Fisher Scientific), according to the manufacturer's protocol. Six hours after transfection, the medium was replaced with fresh cDMEM. Lentiviral supernatants were collected 48 h post-transfection, concentrated and filtered. Viral supernatants were stored at −80 °C until further use.

One day prior transduction, primary human T cells were isolated from healthy donors, seeded at a concentration of $1 \times 10^6$ cells per mL (500 µL per well) in a 48-well plate and activated for 24 h as previously described. Lentiviral supernatants were thawed and mixed thoroughly prior to use. Before virus addition, 333 µL medium was removed from each well, and Polybrene was added at a final concentration of 5 ng/mL. Lentivirus was added at an MOI of 3, followed by centrifugation for 45 min at $800 \times g$ and 32 °C. After centrifugation, plates were incubated for 4 h, after which fresh cRPMI supplemented with 200 IU/mL IL-2 was added to each well. CAR-T cells were expanded as described previously and used within three weeks of generation for in vitro functional analyses.

## Surface antibody staining

48-72 h after transduction, primary T cells or Nur77 Jurkat reporter cells were harvested, washed in 200 µl FACS buffer (1× PBS, 0.5% bovine serum albumin) and evaluated for CAR surface expression by flow cytometry. Up to $5 \times 10^6$ cells were stained in 50 µl mastermix containing all required fluorophore-conjugated antibodies at recommended dilutions (Tab. S2) for 20 min on ice in darkness. For live/dead discrimination, propidium iodide (PI; Life Technologies−1:100) was added during the last 3 min of incubation. Cells were washed twice and filtered with 200 µl FACS buffer prior to acquisition on a Cytoflex S flow cytometer (Beckman Coulter). Data analysis was performed with the FlowJo software v10.8.0 (Treestar).

Where cell quantification was demanded, 123count eBeads (Thermo Fisher Scientific) were added to individual samples according to the manufacturer's protocol. Specific information is supplied with each experiment.

## Fluorescent-activated cell sorting

For cell purification, CAR-engineered PBMCs were harvested and washed with 5-10 mL FACS buffer. Up to $5 \times 10^6$ cells were incubated in 50 µL antibody mastermix for 20 min in the dark at 4 °C and washed at least twice with 10 mL ice-cold FACS buffer. Cell aggregates were removed by passing cells through a sterile 30 µm filter. The whole procedure was performed under sterile conditions. The PI solution was added immediately before cell sorting. CAR-T cells were sorted on a MoFlo Astrios cell sorter (Beckman Coulter) at RT and cultivated afterwards as described above.

## SPR and epitope mapping

SPR experiments were performed using the Biacore T200 instrument (Cytiva). A CM5 sensor chip was pre-treated with 50 mM NaOH

(3×1 min), activated with NHS/EDC for 10 min and coated with 50 µg/mL Strep-Tactin XT (diluted in immobilization buffer, pH 4.5) for 10 min. After immobilization, inactivation of reactive ester groups was conducted by addition of 1 M ethanolamine for 10 min. All reagents originated from the Twin-Strep-tag Capture Kit (IBA Lifesciences) or the amine coupling kit (Cytiva). Twin-Strep-tagged scFv FLEXamers were captured at a concentration of 50 nM at a flow rate of 10 µL/min for 60 s. For analysis of the binding between the different scFvs and CD19, a stability-engineered variant of the CD19-ECD, termed SF-CD19, was used as the soluble interaction partner[44]. SF-CD19 was expressed in HEK293 6E cells, followed by cleavage of the expression tags, as described previously[82]. Different concentrations of SF-CD19 were injected at a flow rate of 30 µL/min in running buffer HBS-EP (Cytiva) at 25 °C, followed by injection of running buffer only. The chip surface was regenerated after each cycle by injection of 3 M GuHCl for 60 s at a flow rate of 10 µL/min. Data were fitted with a 1:1 kinetic binding model using the global data analysis option available within Biacore T200 Evaluation Software (Cytiva).

Competition assays were performed in a similar setup. 50 nM of Twin-Strep-tagged scFv FLEXamers were captured for 60 s with a flow rate of 30 µL/min, followed by injection of 10 nM SF-CD19 for 180 s and a buffer injection for 360 s. The chip surface was regenerated as described before, and 50 nM of Twin-Strep-tagged scFv FLEXamers were injected again for capture on immobilized StrepTactin XT. In a second step, a mixture consisting of 10 nM SF-CD19 and 100 nM FMC63 scFv was injected. Raw binding curves were plotted after subtraction of buffer baseline.

### Tubulin tyrosine ligase (TTL)-mediated functionalization

Conjugation with fluorescent dye was performed as previously described[54]. scFv buffer was exchanged with FPLC buffer (or MES/K if concentrations are < 1.8 mg/mL) by size-exclusion chromatography (Zeba Spin desalting columns, 7 K MWCO) prior quantification. TTL-catalyzed ligation of 3-acid tyrosine to scFv FLEXamer was performed in 100 µL consisting of 20 µM scFv FLEXamers, 5 µM TTL, and 1 mM 3-acid tyrosine in TTL-reaction buffer (20 mM MES, 100 mM KCl, 10 mM MgCl2, 2.5 mM ATP, and 5 mM reduced glutathione) at 25 °C for 3 h. After buffer exchange to 20 mM Tris-HCl and 50 mM NaCl (pH 8) by Zeba Spin columns, azido-FLEXamers were directly functionalized via click chemistry. For that, 20 µM azido-FLEXamers were incubated with 200 µM DBCO-PEG4-Atto-488 at 16 °C for 15 h, followed by buffer exchange to FPLC storage buffer. Functionalization efficacy and concentration were determined by SDS-PAGE. Functionalized proteins were shock-frozen in liquid nitrogen and stored as aliquots at −80 °C for further usage.

### Flow cytometry-based $k_{off}$-rate measurement

To determine anti-CD19 CAR/CD19 dissociation kinetics ($k_{off}$-rate), the flow cytometry-based platform for the measurement of TCR avidity was adapted[53]. Therefore, 0.2 µg Atto488-conjugated scFv molecules were multimerized with 1 µL Strep-Tactin APC in 50 µL FACS buffer for 30 min on ice in the dark. Up to 5 × 10⁶ isolated PBMCs were incubated with 50 µL scFv-StrepTamer for 45 min on ice in the dark. Cells were stained after 25 minutes with anti-CD20 antibody, for target B-cell distinction, and a unique combination of four anti-CD45 antibodies (conjugated with Pacific Blue, Pacific Orange, PerCP, ECD) for multiplexing, enabling the $k_{off}$-rate measurement of up to 16 scFvs at once. Propidium iodide was added during the last 3 min of incubation for live/dead discrimination. Cells were washed twice in FACS buffer and filtered through a nylon mesh for further analysis. Individually stained samples were mixed in a final volume of 2.1 mL FACS buffer and acquired on a Cytoflex S flow cytometer at a flow rate of 130 µL/min under constant cooling at 4 °C or 20 °C. After 40 s of acquisition, 2.1 mL of 2 mM D-Biotin solution was injected, followed by an additional 30 min of acquisition. Baseline mean fluorescent intensities

(MFIs), especially important for CAR scFvs with strong binding strength, were conducted via acquisition of remaining samples stored for 2 h at 4 °C or RT. All measurements were performed in technical triplicates. Dissociation kinetics were analyzed using FlowJo (Treestar). Fluorescence intensities of scFv-Atto488 and the Strep-Tactin-APC of CD45⁺CD20⁺ cells were exported into GraphPad Prism (GraphPad Software). Curve fitting and calculation of dissociation kinetics were performed using a non-linear regression function. The starting value of the curve was set directly at the beginning of the dissociation of Strep-Tactin APC. Baseline values of the scFv-Atto488, which were not reached during the acquisition at 4 °C, were adapted by hand using measured baseline values during the 20 °C measurements.

### Nur77-Jurkat reporter cell assay for antigen-specific activation

To evaluate antigen-specific activation, 5 × 10⁴ CAR-engineered Nur77-tdTomato Jurkat cells were co-cultured with 5 × 10⁴ CD19⁺ Raji-ffluc-GFP target cells in 200 µL RPMI into a 96-well U-bottom plate at 37 °C. Effector cell numbers were determined based on the expression of the transduction marker EGFRt⁺. While CAR-transduced Nur77 Jurkat cells without target cells served as a negative control, Phorbol-mystriate-acetate (PMA−25 ng/mL) and ionomycin (1 µg/mL) (Sigma Aldrich) were used as positive controls. After 3 h of incubation, cells were harvested and stained with anti-EGFR antibody and Streptavidin. Reporter gene activation was analyzed by frequency analysis of tdTomato of EGFRt⁺GFP⁻ lymphocytes. Experiments were performed in technical duplicates and biological triplicates.

### Intracellular cytokine staining

Primary CAR-T cells were rested for two days in cRPMI after interleukin stimuli. 5 × 10⁴ EGFRt⁺ CAR-T cells were co-cultured with 5 × 10⁴ (E:T 1:1) or 12.5 × 10³ (E:T 1:0.25) CD19⁺ Raji-ffluc-GFP target cells in 200 µL cRPMI into a 96-well U-bottom plate at 37 °C. CAR-T cells in medium and PMA (25 ng/mL) supplemented with Ionomycin (1 µg/ml) were used as negative and positive controls, respectively. After 1 h, cytokine secretion was stopped by addition of 50 µL Brefeldin A ('Golgi Plug'− BD Biosciences) at 1 µg/mL. After an additional 4 h, cells were harvested and incubated with ethidium-monoazide-bromide (EMA; Life Technologies, 1:1000) for 10 min on ice, exposed to bright light for live/dead discrimination. Surface staining (CD8, EGFRt, Strep-tag) was performed as described before at 4 °C in the dark for 20 min. After washing in 200 µL FACS buffer twice, cells were fixed and permeabilized with Cytofix/Cytoperm (BD Biosciences) at 4 °C for 20 min, according to manufacturer's instructions. The samples were washed twice with PermWash buffer (BD Biosciences), cells were stained intracellularly with anti-IFN-γ, anti-TNF-α and anti-IL-2 antibodies on ice in the dark for 20 min (Tab. S2), followed by two additional washing steps in PermWash buffer and filtered into 200 µL FACS buffer for analysis on a flow cytometer. Experiments were performed in technical and biological triplicates.

### In vitro killing assay

A 96-well E-Plate was filled with 50 µL DMEM and equilibrated in the xCelligence RTCA MP Real-Time Cell Analyzer (ACEA Bioscience) for 30 min at 37 °C to determine the baseline impedance signal. 100 µL of CD19⁺ and CD19⁻ adherent HEK293T in cDMEM (1.5 × 10⁴ cells) or ROR1⁺ A549 ffluc cells in cRPMI (1.0 × 10⁴ cells) were added and incubated for 24 h. Impedance measurements were conducted every 15 min. 100 µL cDMEM or cRPMI were aspirated and replaced by 100 µL containing 1.5 × 104 (1:1 ratio) or 7.5 × 104 (4:1 ratio) EGFRt⁺ CAR-T cells. In separate experiments, 100 µL containing 4.0 × 10⁴ (4:1 ratio), 2.0 × 10⁴ (2:1 ratio), 1.0 × 10⁴ (1:1 ratio) or 0.5 × 10⁴ (0.5:1 ratio) EGFRt⁺ CAR-T cells were added to ROR1⁺ A549 ffluc cells were added. 2 % Triton-X100 solution (Biorad) and cRPMI medium without effector CAR-T served as positive and negative controls, respectively. Effector and target cells were co-incubated for 48−72 h with measurement

intervals of 15 min. Cytotoxic kinetics were analyzed using the software RTCA xCelligence v2.0 (ACEA Bioscience). Cell indices were normalized to the last measurement before addition of CAR-T cells. The samples were measured in technical triplicates and biological duplicates.

## Flow cytometry-based cytotoxic CAR-T-cell assay

Additionally, cytotoxic effector function of CAR-T cells was determined in a co-culturing approach with CD19⁺ Raji-ffluc-GFP or CD19⁺ Nalm6-ffluc-GFP (expressing either wildtype or low antigen level) target cells. The reduction of GPF+ target cells after 24 h and 48 h served as a flow cytometry-based readout. Therefore, $2 \times 10^5$ target cells were plated in 50 μL in a 96-well U-bottom plate. Frequencies and cell numbers of EGFRt⁺ CAR-transduced T cells were determined as described before. Serial dilution of CAR-T cells was performed, and 150 μL with either $8 \times 10^4$, $2 \times 10^4$, $1 \times 10^4$, $0.5 \times 10^4$ or $0.25 \times 10^4$ EGFRt⁺ CAR-T cells were added to the target cells (E:T = 4:1, 1:1, 0.5:1, 0.25:1, 0.125:1), respectively. Non-CAR-transduced T cells (mock) derived from the same donor and target cells without T cells served as negative controls. After 24 h or 48 h, cells were harvested. 123countTM eBeads were added to extrapolate absolute cell numbers. The samples were stained with dye-conjugated anti-EGFR and anti-CD8 antibodies, as well as Streptavidin, as described before.

Target cells were discriminated by internal expression of GFP. All samples were measured in triplicate. The percentage (%) of target cell killing was calculated based on the absolute number of target cells in the mock condition. That absolute number of mock T cells was set as 100%, and the number of targets in the following T-cell conditions was expressed as a percentage of survival. The elimination index was calculated as follows: 1−(number of residual target cells in the presence of target antigen-specific CAR-T cells/ number of residual target cells in the presence of mock T cells).

## Chronic antigen stimulation assay

To investigate the functionality of a combined-affinity CAR-T-cell product in a chronic antigen setup, $3 \times 10^5$ CAR-T cells were co-cultured with $1.2 \times 10^6$ Nalm-6-ffluc-GFP or Jeko-1 ffluc-GFP cells in a 24-well plate format. Every second day, 100 μL cells were collected and stained as described above. Tumor elimination and CAR-T-cell expansion were evaluated based on 123count eBeads. Tumor cells were added to restore an E:T ratio corresponding to 1:4 according to absolute cell numbers. The described phenotype was observed after a minimum of four rounds of stimulation.

## In vitro CRS evaluation

For the evaluation of CRS side effects in vitro, the protocol of Nouri et al. was adapted[83]. Therefore, $2 \times 10^5$ CAR-T cells were co-cultured with $1 \times 10^5$ CD19-expressing K562 cells and $1 \times 10^5$ monocytes for 24 h in IMDM medium (Sigma Aldrich) supplemented with 50 mL FBS and 5 mL penicillin-streptomycin. Monocytes were isolated from freshly thawed PBMCs derived from the same donor used for generating CAR-T cells through the MojoSort human CD14 monocyte isolation kit according to manufacturer's protocol (Biolegend). Wildtype K562 cells served as the negative control. After 24 h, the supernatant was collected and stored at −20 °C until further usage. For the detection of CRS-associated cytokines, the ELISA MAX Deluxe Set Human IL-6 Kit (Biolegend) or the human essential immune response LegendPlex kit (Biolegend) was used according to the manufacturer's protocol. Supernatant was diluted at least 1:20 with IMDM medium before cytokine concentrations were determined. Experiments were conducted in technical and biological triplicates.

## Mouse experiments

NSG-SGM3 (NOD.Cg-Prkdcscid Il2rgtm1Wjl Tg(CMV IL3,CSF2,KITLG) 1Eav/MloySzJ; # 013062) mice (male and female, 8-11 weeks old) were acquired from The Jackson Laboratory or derived from in-house breeding. All animals were group-housed under specific pathogen-free conditions at controlled ambient temperature of 20 °C at our mouse facility at the Technical University Munich, Institute for Medical Microbiology, Immunology and Hygiene. A 12-hour light/dark cycle was maintained to regulate the circadian rhythm, and a relative humidity of 50−60%. The conducted animal experiments were approved by the local authorities especially the district government of Upper Bavaria (Department 5−Environment, Health and Consumer Protection ROB-55.2-2532.Vet_02-18-162).

To investigate the anti-leukemic efficacy of CAR-T cells with different antigen-binding affinities, NSG-SGM3 mice were challenged with $5 \times 10^5$ Raji-ffluc-GFP or Nalm-6-ffluc-GFP cells intravenously (i.v.). After 7 days, CAR-T cells were i.v. injected in bulk or after purification. For bulk injection, CAR-T-cell counts were calculated based on the frequency of EGFRt⁺ cells and adjusted accordingly. Independent mouse experiments were conducted using CAR-T cells generated from PBMCs from different donors. Tumor progression was analyzed by bioluminescence imaging weekly. Therefore, mice were intraperitoneally injected with 150 mg/kg XenoLight D-Luciferin Potassium Salt dissolved in PBS (PerkinElmer), followed by anesthesia with 2.5 % isoflurane in the RAS-4 Rodent Anesthesia system (PerkinElmer) and imaged with the IVIS Lumina Imaging System (PerkinElmer). The analysis was conducted by quantification of photon/sec/cm²/sr with the Living Image 4.5 software (PerkinElmer). CAR-T-cell expansion was evaluated by blood staining once a week. Therefore, 50−100 μL blood was collected, erythrocytes were lysed with 10 mL ACK buffer (15 M Ammonium chloride−Sigma Aldrich, 17 mM Tris-HCl−Roth) and stained accordingly. Mice were euthanized by cervical dislocation upon reaching predefined human endpoints or at the completion of the experiment, and bone marrow and spleen were subsequently harvested and analyzed 30 days after CAR-T-cell transfer. Bone marrow was isolated from femur and tibia and incubated with 3 mL ACK for 3 min. Spleens were mashed through a 100 μm cell sieve and treated with 5 mL ACK buffer for 5 min. Flow cytometry staining was performed as mentioned before. For affinity combination experiments, CAR transduction efficacies were determined based on the transduction marker EGFRt. Individual CAR proportions of the product after combination were evaluated based on the expression of fluorescent proteins as described above.

For the in vivo assessment of CAR-associated side effects, 3−4-week-old female NSG-SGM3 mice were sublethally irradiated with 1 Gy and i.v. injected with $2 \times 10^4$ CD34⁺ HSCs on the following day. After 8-12 weeks, the reconstitution of the human immune compartment was confirmed via blood analysis and mice were challenged with $5 \times 10^5$ Raji-ffluc-GFP followed by CAR-T-cell administration as described above. CRS intensity was evaluated based on the body weight loss, changes in temperature development and serum cytokine levels. To evaluate the concentration of inflammatory cytokines in the circulation, serum samples were collected 3 and 5 days after CAR-T-cell transfer and analyzed with the capture-bead-based LegendPlex Kit (Biolegend) according to the manufacturer's protocol. Data was analyzed with the LEGENDplex Qognit, Inc. Software (Biolegend).

## Single-cell RNA sequencing

CAR-T cells were isolated from bone marrow of NSG-SGM3 30 days after transfer. Bone marrow was harvested as described above. Up to $1 \times 10^7$ cells were stained in 100 μL antibody mix as described above. In parallel, each sample was labeled with a unique Hashtag antibody (TotalSeq™ B Barcodes Biolegend) for the following demultiplexing (Tab. S3). 1000 CAR-T cells per donor mouse were sorted for GFP⁻, CD19⁻, EGFRt⁺. After sorting, the supernatant was removed, and the cells were resuspended in Mastermix + 37.8 μl of

water before 70 μl of the cell suspension was transferred to the Chromium next GEM chip.

The integrity of the cell pellet was checked under the microscope frequently before loading the sample on the chip. 10X experiment was conducted according to the manufacturer's protocol (Chromium next GEM Single Cell 3' v3.1 with dual index). Sample quality was confirmed with a High-sensitivity DNA Kit on a Bioanalyzer 2100, and libraries were quantified with the Qubit dsDNA hs assay kit. All steps were performed with RPT filter tips and LowBind tubes.

### scRNA-seq data analysis

The human reference GRCh38-2024-A, was used for Transcriptome annotation (CellRanger 7.0.0). To identify CAR transcripts in single cells, the complete transcript of the CAR constructs (JCAR017, JCAR021) was added to the reference genome. Data preprocessing has been performed according to the current best practice in scRNA sequencing analysis[84,85]. Data analysis was performed with SCANPY V1.9.1. In brief, single cells were annotated by their respective CAR transcripts (JCAR017, JCAR021); only cells with a unique identification of a CAR construct were annotated as JCAR017 or JCAR021, respectively. Subsequently, cells with fewer than 200 genes, as well as genes present in fewer than 20 cells were excluded. Counts were normalized per cell, logarithmized, and the variance was scaled to unit variance and zero mean. The number of counts was regressed out before highly variable genes were identified and filtered. DNA-Barcode demultiplexing was performed with HashSolo included in SCANPY. Differential gene expression analysis was performed using SCANPY and diffxpy (https://diffxpy.readthedocs.io/en/latest/index.html). Gene ontology analysis was performed using g:Profiler toolkit[86]. Gene set enrichment analysis was performed using GSEAPY[87].

### Reporting summary

Further information on research design is available in the Nature Portfolio Reporting Summary linked to this article.

## Data availability

The single-cell RNA sequencing data generated in this study have been deposited in Gene Expression Omnibus under accession number GSE322706. Processed single-cell datasets and all source data supporting the findings of this study have been deposited in the Zenodo repository (https://doi.org/10.5281/zenodo.18861476). Source data are provided with this paper.

## Code availability

Scripts supporting the data were based on existing computational tools. The code used for the analysis is available at GitHub https://github.com/Busch-Lab/2026_Warmuth_Doetsch_affinity-combined-CAR-T_cell-products_scRNAseq and archived at Zenodo (https://doi.org/10.5281/zenodo.18861476).

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

## Acknowledgements

We thank the members of the Busch laboratory at MIH for helpful suggestions and technical assistance. We especially thank Immanuel Andrä, Katharina Hofmann, Tanja Roßmann-Bloeck and Corinne Angerpointner for cell sorting assistance. This work was funded by the Deutsche Forschungsgemeinschaft (DFG, German Research Foundation) SFB-TRR 338/1 2021-452881907 (projects A01, A02, A05) and SFB 1371 395357507 (project P04). D.H.B., L.W., M.P.P. and M.C. also received funding from the Innovative Medicines Initiative (IMI) 2 Joint Undertaking under grant agreement No. 945393 (T$^2$EVOLVE). The contribution of K.M. was supported by GRK2668 (ID 435874434; project-B01). E.D. was supported by the Deutsche Zentrum für Infektionsforschung (DZIF, TTU 07.713). J.S. and M.W.T. were supported by the Austrian Science Fund (FWF Project W1224 – Doctoral Program on Biomolecular Technology of Proteins – BioToP). The SPR equipment was kindly provided by the Connective Base GmbH and the project was supported by the BOKU Core Facility Biomolecular & Cellular Analysis. M.H. received funding from the German Cancer Aid, Pre-clinical Drug Development Program - preCDD, CAR FACTORY (Project Number 70115200). The contribution of J.M. was supported by the Bayerische Transformations- und Forschungsstiftung (Baycellator).

## Author contributions

S.D., L.W., M.P.P., E.D. and D.H.B. designed research; S.D., L.W., R.V.M., K.M., M.T., A.H., M.H., S.B., S.E., C.S. and J.M.S. conducted experiments, S.D. and L.W. analyzed the data, A.S. and S.W. analyzed single-cell RNA sequencing data, G.S. provided umbilical cord blood for HSC isolation, J.S., C.U.Z. and M.W.T. performed and analyzed SPR measurements, J.M.S. and M.C. provided critical feedback to data interpretation and establishment of the humanized mouse model, J.M., T.N. and M.H. provided resources; S.R.R. provided cell lines and critical feedback, L.W., M.P.P., E.D. and D.H.B. wrote the manuscript with the support of all authors.

## Funding

## Competing interests

D.H.B. and M.P.P. are co-founders of Match Medicines GmbH. L.W., S.D., E.D, M.P.P., and D.H.B filed a patent application related to the affinity combination strategy presented in this manuscript. S.R.R. has received research funding from Juno Therapeutics, a Bristol Myers Squibb company and Lyell Immunopharma; has rights to royalties from Juno Therapeutics, a Bristol Myers Squibb company and Lyell Immunopharma; has served as a consultant for Lyell Immunopharma, Adaptive Biotechnologies and Outpace Biosciences; has patents licensed to Juno Therapeutics, a Bristol Myers Squibb company, and Lyell Immunopharma; and has stock or stock options from Lyell Immunopharma, Adaptive Biotechnologies, and Outpace Biosciences. T.N. and M.H. are listed as inventors on patent applications and granted patents related to CAR-T technologies that have been filed by the Fred Hutchinson Cancer Research Center, Seattle, WA (M.H.) and the University of Würzburg, Würzburg, Germany (T.N., M.H.) and that have been, in part, licensed by industry. M.H. is co-founder and equity owner of T-CURX GmbH, Würzburg, Germany. M.H. received speaker honoraria from BMS, Janssen, and Kite/Gilead. The remaining authors declare no competing interests.

## Additional information

[1]Institute for Medical Microbiology, Immunology and Hygiene, Technical University of Munich, Munich, Germany. [2]Graduate Center of Medicine and Health, TUM School of Medicine and Health, Technical University of Munich, Munich, Germany. [3]Institute of Biochemistry, Department of Natural Sciences and Sustainable Resources, BOKU University, Vienna, Austria. [4]Medical University of Vienna, Department of Transfusion Medicine and Cell Therapy, Vienna, Austria. [5]Clinic and Polyclinic for Gynecology, TUM University Hospital, TUM School of Medicine, Technical University of Munich, Munich, Germany. [6]Chair for Cellular Immunotherapy, Department of Medicine II, University Hospital Würzburg, Würzburg, Germany. [7]Fraunhofer Institute for Cell Therapy and Immunology (IZI), Leipzig & Branch Site Cellular Immunotherapy, Würzburg, Germany. [8]National Center for Tumor Diseases (NCT), Site Würzburg-Erlangen-Regensburg-Augsburg (WERA, Würzburg, Germany. [9]Bavarian Center for Cancer Research (BZKF), Lighthouse Cellular Immunotherapies, Würzburg, Germany. [10]Innovative Immunotherapies Unit, Division of Immunology, Transplantation, and Infectious Diseases, IRCCS San Raffaele Scientific Institute, Milan, Italy. [11]Translational Sciences and Therapeutics Division, Fred Hutchinson Cancer Center, Seattle, WA, USA. [12]German Center for Infection Research (DZIF), partner site, Munich, Germany. [13]These authors contributed equally: Linda Warmuth, Sarah Dötsch. [14]These authors jointly supervised this work: Dirk H. Busch, Elvira D'Ippolito. ✉e-mail: dirk.busch@tum.de; elvira.dippolito@tum.de

