## [Transparent Peer Review file · Nature Communications]

Balancing the efficacy and safety of Chimeric Antigen Receptor T cell therapy by affinity combination

Corresponding Author: Professor Dirk Busch

Version 0:

Reviewer comments:

Reviewer #1

(Remarks to the Author)

Balancing the efficacy and safety of Chimeric Antigen Receptor T cell therapy by affinity combination

In this manuscript Warmuth, Dotsch, et.al, propose the novel concept that a combination therapy using anti-CD19 CARs with different affinities (2.7 nM and 140 nM) could achieve comparable efficacy to high affinity CARs alone, while lowering off-tumor toxicity. The authors suggest this strategy has the potential to lower the risk of cytokine release syndrome (CRS) and immune effector cell-associated neurotoxicity syndrome (ICANS), thereby expanding the therapeutic window of CAR T cells. The authors support their hypothesis through a series of in vitro and in vivo assays, including studies in NSG and a humanized mouse model, to assess both efficacy and toxicity.

A particularly notable finding is that high affinity CARs in the bone marrow exhibit a distinct phenotype in the presence of low affinity CARs as determined by scRNA-seq. The authors suggest a synergistic interaction between the low- and high-affinity CARs, enabling a balance between therapeutic efficacy and safety.

The work is relevant to the field of CAR T cell immunotherapy, where achieving an optimal balance between efficacy and toxicity remains a major challenge. The proposed affinity-combination approach should be of interest to the immunotherapy community. However, the scope of the study is currently limited to CD19-directed CARs with a specific architecture (a CD28 transmembrane domain, a 41BB co-stimulatory domain and a CD3z signaling domain) which may constrain the broader applicability of the findings. While it is reasonable for the authors to stay within this defined system, expanding the study to include additional target antigens and/or other CAR architectures would substantially increase the impact of the approach.

The authors present affinity measurements for a well-established high affinity murine ScFv (JCAR017 based on FMC63 - 2.7 nM) and a lower affinity humanized version (JCAR 021 - 140 nM). They then developed a collection of affinity variants by mutating the JCAR021 scFv and incorporating these variants into synthetic CAR constructs to evaluate performance. While this provides a careful and systematic evaluation of affinity-dependent CAR function, similar studies have been previously reported. For example:

He et. al (Science Immunology, 2023) characterized several affinity variants of FMC63, including WT - 4.5 nM, Y261A - 682.5 nM, Y70A - 275 nM and showed functional differences in CAR T cells depending on affinity.

<https://www.science.org/doi/10.1126/sciimmunol.adf1426>

Barden et al (Journal for Immunotherapy of Cancer, 2024) and Liu et al have explored the relationship between ScFv affinity and CAR efficacy in other tumor antigens, such as HER2, with conclusions that moderate affinity cAR may outperform both high and low affinity variants under certain conditions.

<https://jitc.bmj.com/content/12/12/e010208>

Given this context, the novelty of the current work lies not in the generation of affinity variants but in the combination of two CARs of differing affinity to enhance the therapeutic index. This concept is interesting and underexplored.

However, to strengthen the affinity exploration, the authors might consider including in vitro killing assays using CD19+ targets of different densities at lower E:T. Previous work has shown in vitro differences that are affinity dependent (Fig. 4C - reference above). Under the current conditions, where T cell numbers are high, such differences might be masked.

The overall conclusions of the manuscript are promising. However, several key claims, particularly those regarding the mechanism of synergy or communication between high- and low-affinity CAR T cells in the combined product, would benefit from additional clarification or supporting evidence.

Specifically

Efficiency of in vitro killing across a broad affinity spectrum of CAR binders

Current E:T ratios used to explore differential in vitro killing across variants (Fig. 2g) show early saturation in target killing for most variants. Most variants that exhibit intermediate behavior between the control curve and JCAR021 express low levels of CAR or do not express it at all (Suppl. Fig. 3d). More challenging E:T ratios may be able to demonstrate dependence of killing dynamics on CAR affinity. Similarly, exploring how variants with different affinities kill targets with low antigen density would provide another way to test differences in vitro. The authors already demonstrated a relationship between CAR affinity and capacity to kill low antigen density targets (Fig. 5n). Exploring affinity variants across these two axes (total antigen by E:T ratio or using low antigen density per cell) could help validate the claim that there are no functional differences across the broad spectrum of CAR affinities.

CAR T cell toxicity associated with affinity.

Despite generating several affinity variants, the authors only explored the original 2 CARs based on JCAR017 and JCAR021. Although there is a clear correlation between CAR affinity, dose, and toxicity, it would have been interesting to observe variants with intermediate affinities between these two (e.g., S168C, S227K, S50A). This is particularly relevant because Fig. 4b and 4h suggest that for JCAR017, a dose between 1M and 4M is sufficient to control tumors without significant toxicity; however, not even 8M JCAR021 can control tumors.

Ratio of low/high affinity CAR T cells over time in vivo vs in vitro -

The dynamics of CAR T cell populations in the 50:50 combination differ substantially between the in vitro and in vivo settings. In vitro (Fig. 5g-i), the total expansion resembles the high-affinity CAR alone, but the population eventually shifts toward low-affinity CAR dominance (~30:70 by day 28, Fig. 5i). In contrast, in vivo data from day 10 post-ACT (Supp. Fig. 12c, left) show similar cell numbers for both the low-affinity and combined products across different dose levels, which suggests different expansion or persistence dynamics. It would strengthen the manuscript if the authors could comment on these discrepancies and discuss possible biological or experimental reasons behind them.

The authors propose that the high-affinity CAR T cells are protected from exhaustion in the combination setting. However, the absolute numbers of high-affinity CAR T cells (JCAR017) in both the solo and combination treatments appear to decline similarly over time (Supp. Fig. 12c, middle). Would earlier time points for the high-affinity CAR alone show a more comparable phenotype to that observed in the combination setting? Alternatively, could the dominant expansion of the low-affinity CAR population reduce the number of high-affinity CAR interactions with target cells, thereby indirectly preserving their phenotype? It would be useful if authors split the analysis of the combined JCAR021 and JCAR017 CARs for each individual treatment group (0.8/0.8 and 0.2/1.4). These two groups have different tumor control profiles (Fig. 6c) and total number of cells (Suppl. Fig. 12d). It would be interesting to see if the described phenotypical differences are maintained when comparing JCAR017 alone with JCAR017 in the 50:50 ratio.

The current data set lacks an additional control using high-affinity CAR T cells with untransduced T cells or CARs of unrelated specificity at matched total cell doses. This limits the ability to conclusively attribute phenotypic shifts in the high-affinity CAR population to active synergy or indirect effects from the low-affinity CAR T cells. Including such controls would help determine whether the observed effects are specific to affinity combinations or merely a consequence of changes in T cell dose or competition.

It remains unclear whether the differences in CAR T cell phenotype (especially exhaustion profiles) could be influenced by differential cytokine production. Is there in vivo data on cytokine expression profiles comparing high- and low-affinity CARs, or in the combination product? This could help determine whether secreted factors from the low-affinity CAR T cells contribute to shaping the phenotype or function of their high-affinity counterparts.

Several issues could be clarified and revised to improve data interpretation. Addressing the points below would enhance clarity

Replacing the labels "JCAR017" with "High affinity CAR (ScFv - 2.7 nM)" and "JCAR 021" with "Low affinity CAR (ScFv 140 nM)" throughout the figures would make the data more accessible and easier to follow for a broader readership, particularly for those less familiar with the names/codes.

CAT CAR T cells are only explored in Fig. 1. Although it is important that the authors benchmark their findings against other previously described CARs, this could be moved to the supplementary figures, allowing the main figures to focus only on the high and low affinity CARs that are explored further (JCAR017 and JCAR021).

Fig.1g axis label reads E:T 1:4 but the Methods section for the assay description states E:T 1:0.25, which one is it?

Intuitively, one would expect a higher fraction of cells secreting cytokines at higher target numbers. The current axis labels suggest otherwise. Please, clarify

Fig. 1h More E:T conditions and antigen-densities for each construct could reveal differences in the in vitro killing behavior, particularly under suboptimal T cell conditions where functional differences are more likely to emerge (see comment in previous section)

On page 6, the phrase "the individual label of CAR-T cells with fluorescence conjugated anti-CD45 antibodies enabled multiplexing..." seems to indicate that CAR-T cells were generated and stained, while the methods section states that target B cells were stained with anti-CD45 antibodies. Please clarify this discrepancy.

The logic of Fig. 2 is currently somewhat confusing, as it first addresses and classifies CAR T cells based on function, and only later explains that many of the low-functioning CARs do not even express the receptor as a surface protein. One suggestion for reorganizing the figure is to first establish the affinity of the variants, then describe their expression as CARs in primary T cells, and only proceed with the functional assays for those that show clear CAR expression (based on EGFR^{T+} and STII⁺). Additionally, binning the CARs into groups does not allow the reader to appreciate interesting cases of function versus affinity. The heat map in Fig. 2i is a good summary, but including some XY plots of affinity versus target lysis or affinity versus IL-2 secretion could help to highlight interesting cases. That would also help to feature variants that are used in following figures, allowing a better interpretation of those results.

Fig.4 can be rearranged to improve clarity. Figure panels that showed immune reconstitution (Suppl Fig.7 c and d) could be moved to main Fig.4. Also, dose titration is interesting mostly only in the context of affinity variants. Panels Fig.4 b to g can be moved to supplementary, allowing a clearer focus to Panels Fig4h to k. Some panels from Suppl Fig 9 can be then moved also to the main Fig. 4.

Which cell line is used in Fig.4 experiments (Raji or Nalm-6), please specify in figure's legend

Mapping the mutations in the JCAR021 ScFv onto its structure and color-coding them by functional impact would add valuable insight. This visualization could highlight whether key residues are localized near the CD19 binding interface or whether functional "hotspots" exist in certain regions of the antibody. This may also help clarify why some affinity variants behave differently than expected.

Fig. 5 would benefit from including data for all the variants mentioned in the figure. The monocyte cytokine release assay and the chronic stimulation assay should be performed with all the described variants, as well as their combinations. Additionally, at least one variant with intermediate affinity between JCAR017 and JCAR021 should be tested (e.g., S168C, S227K, S50A). This would allow for better interpretation of the role of affinity in toxicity and exhaustion prevention.

Similarly, antigen density is an important factor to evaluate, as it may reveal differences in target killing by the various affinity variants, as observed in Fig. 5n. In addition, Fig. 5n shows that target killing by the combination of CAR T cells appears to reach saturation at 24 hours, which does not allow for a proper evaluation of whether the effect of the combined CAR T cell approach is driven by phenotypic differences in the high-affinity CAR. Testing different E:T ratios could help the reader observe these potential differences, such as instances where the same amount of high-affinity CAR fails to eliminate tumor cells while the combination, even at a lower high-affinity CAR dose, is effective. Furthermore, testing similar ratios of JCAR017 and JCAR021 alongside the other variants would also be informative.

Figure 6e. Why does there seem to be much less cells for both high- and low-affinity CAR T a in the UMAP in the 50:50 condition than in the 90:10 condition? The absolute counts in S12c show otherwise, could you provide an explanation between the count trends in the blood and bone marrow? Similarly, as described before it would be useful if authors split the analysis of the combined JCAR021 and JCAR017 CARs for each individual treatment group (0.8/0.8 and 0.2/1.4). Also, including it would be useful for the interpretation of the results to include in the comparisons in Fig. 6h an j the profile of JCAR021 alone and in combination.

The authors claim: "Our data show that lower-affinity CAR-T cells remain functionally active in the presence of high-affinity CAR-T cells, leading to a more controlled and sustained immune response." However, this claim is difficult to support with the current experiments. Although it can be assumed that lower quantities of JCAR017 require the presence of JCAR021 to achieve similar survival (Fig. 6b), effective tumor control appears to require at least 0.8 million high-affinity CAR-T cells to prevent tumor growth (Fig.6c). Additionally, low-affinity CAR-T cells achieve similar numbers in the bone marrow under conditions that differ in terms of tumor control (i.e., 1.6 million single low-affinity cells versus 0.8 million in combination) (Fig.6d). Including an analysis of JCAR021 from the scRNAseq data, as previously suggested, may help to substantiate this claim.

Please clarify if the same donor T cells were used throughout the manuscript or different ones. If a single one, adding other donor T cells would strengthen the results.

Methods are mostly adequate. Extra details in some sections would help to reproduce authors' work. Specifically:

In figure legends, specify the number of donors used in the experiments.

Add source of retroviral expression vector (Addgene?, other lab?)

Which number of RD114 that was seeded, in which plate/flask format, for how long before transfection.

In which media transfection was performed, and was media exchanged during those 72h?

For how long the transduced primary T cells were maintained in culture before CAR expression was analyzed. How long after the transduction they were used in in vitro experiments. How long for in vivo experiments.

For which experiments CAR T cells were sorted? For which experiments they were not sorted. Please add this information to the figure legend, particularly for the in vivo experiments.

For the chronic restimulation assay, please describe the plate format used. Also, add how many rounds of stimulation were performed.

For BM and spleen preps, do authors intend to write ACK buffer instead of "ACT buffer"?

Rogelio Hernandez-Lopez

Daniel Hoces Burga

Reviewer #2

(Remarks to the Author)

Warmuth et al. present interesting results showing that the combination of high- and low-affinity CARs targeting Cd19 antigen can balance the antitumor response by reducing toxicities without compromising efficacy. The authors generated and thoroughly tested a panel of CD19 binders and compared them to clinically used CAT. The authors evaluated T cells with different binders alone or in combination in vitro and in vivo, including in a humanized model to assess CRS. Overall, the manuscript is well presented, original, and addresses a critical question in the field. My only concern is that the study would benefit from deeper mechanistic analyses to determine how low-affinity CARs mitigate high-affinity CAR exhaustion and excessive stimulation. Is this achieved through transient antigen masking, reduced tumor burden, or potential cytokine secretion? Addressing this question would further strengthen the authors conclusions. A minor point that should be addressed in the limitations section is the challenge of translating these findings to patients, given the heterogeneity both between patients and across different antigens, a limitation noted by the authors in the discussion.

Reviewer #3

(Remarks to the Author)

The presented manuscript elaborates the impact of receptor-affinity towards efficacy and safety of CD19 targeting CAR T cells. In short, the authors describe that in vivo, but not in vitro function correlates with affinity. Using improved murine CRS models, the authors show that in high, but not in low affinity CAR Ts, toxicity is correlated to dose. In a last step, the authors evaluate the combination of high and low affinity CAR T cells with the aim to maintain efficacy by reducing toxicities such as CRS.

For long time, CARs were preferentially generated from high affinity antibodies, whereas nowadays, scientists are becoming aware that sometimes, lower affinity might be beneficial in specific indications (e.g. Olson et al, Leukemia, 2022; Flugel et al, Nat Rev Clin Onc, 2023). Thus, the combinatory approach presented here is of significant interest to the field and displays a concept that has not been presented yet.

Most of the conclusions and claims are sufficiently supported, however, due to the large amount of data (varying dosages and combinations) in the murine studies, the reader would benefit from further explanation how studies are connected to each other, and if and how conclusions from one experiment might have shaped the setup of the following study. From the data presented, it looks like 2 Mio JCAR017 are typically curative in the used models, whereas 1Mio JCAR017 is not sufficient to completely eradicate the tumor. Regarding the combination of JCAR017 and JCAR021, there seem to be different ratios that might be beneficiary, and a final conclusion on the optimal ratio is missing. Thus, a final experiment is missing to fully support the main message of the manuscript: Within the CRS model, compare an optimal combination (e.g. 1Mio +1Mio, if that is the chosen dosage) against I) a comparable dose (total cell amount =2 Mio) of high affinity CAR and II) same amount of high affinity CAR as in the combination (1Mio) to make a final decision on the dose as well as to have supportive data on efficacy and toxicity, preferably over long term to address toxicity as well as long term efficacy and potential relapse.

Statistics should only be calculated if $n \geq 3$ and biological (but not technical) replicates are available. Smaller datasets can still be presented but should not be included in statistical analysis.

Besides the above mentioned general comments, there are some details that should be addressed:

Introduction:

- Besides JCAR017 being the "central component of most commercially approved anti CD19 CAR T products" it should be mentioned that also JCAR021 is being evaluated in clinical trials and results have been reported (PMID 39820359 by Gauthier et al, 2025).

Figures:

- Fig. 1h: The wording of figure should be revised: "data was obtained from one experiment".

- Fig 2f: How are constructs assigned to group mfunc and mfunc low? If this assignment is based on function, there is one construct within the CARmlowfunc group that shows cytokine production after antigen spec. stimulation. Please clarify allocation.
- Fig. 2h: In addition to the transduction rate, is the MFI also reduced in the mlowfunc group?
- Fig 3f/S5b: BLI data from the one mouse surviving in the L237W group (d28) is shown in S5b, but not in 3f. Please justify why this has been excluded from the analysis.
- Fig. 3g ff: What is the baseline of the Mock T cells? Were any Mock T cells detectable in blood, BM or spleen?
- Fig S6: Please include average radiance analysis of the remaining mice of d24 as the final day of the experiment
- Fig. 4: How has the murine CRS model been improved? Please indicate for each experiment which cell line was actually used (Nalm-6 or Raji).
- Fig 4h-k: Even though Fig 4 is meant to demonstrate toxicity, the BLI imaging suggests that the low affinity CAR might have an antitumor effect at high dosage while showing a lower toxicity. Is there data available on longer observation periods regarding efficacy?
- Fig: 5b: Does the ametrine fluorescent protein influence CAR expression or functionality?
- Fig 5c: Please clarify the composition of CAR017 and JCAR021 at t=0
- Fig 5a-e: From the presented data, it is not clear if the combination has a benefit at all. A control condition containing the reduced amount of JCAR017 without the addition of JCAR021 would be helpful to fully interpret the results.
- Fig.5n: please include a condition of 0.01×10^6 JCAR017 to enable a conclusion if similar lysis would also be achieved by the lower dose of JCAR017 in the complete absence of JCAR021. In the current data set, Figs. 5 and S10 do not sufficiently support the main message of the paper, as relevant controls are missing.
- Fig. S10: same as Fig 5. In Figure S7, it is shown that giving 2 or 4 Mio of JCAR017 does not have a significant difference in efficacy. Thus, the data from S7 suggest that the observed effect is caused by the reduction of the amount of JCAR017, but not from the combination with the low affinity CAR.
- Fig. S11: in contrast to the text (p.12), Fig. S11 doesn't demonstrate increased safety, but efficacy of combined CAR T. Indeed, a comparison of this "optimal combination approach" to standard dosages with regards to safety and efficacy would strongly support the manuscript.
- Fig. S11a: 1M JCAR017 and 1M JCAR017 +1M JCAR021 are not shown in b/c. if these conditions were not assessed here, but in a separate experiment (d/e), please delete in the experimental set up (a).
- Fig. S11d/e is mentioned in the text in correspondence with Fig S12, maybe consider relocation.

Discussion:

- The affinity variants of the CAR all target the same epitope within CD19. As one suggested mode of action is the competitive binding and availability of antigen binding sites, do you think the effects would be similar when targeting different epitopes?

Reviewer #4

(Remarks to the Author)

Version 1:

Reviewer comments:

Reviewer #1

(Remarks to the Author)

The authors have significantly improved their manuscript. Also they have responded adequately to the questions and comments. We recommend that this article is accepted for publication.

Minor typo in page 12 of the revised manuscript, "JCAR012" perhaps the authors meant JCAR021

Rogelio A. Hernandez-Lopez
Daniel Hoces

Reviewer #2

(Remarks to the Author)

The authors addressed my concerns and significantly improved the manuscript

Reviewer #3

(Remarks to the Author)

Thanks to the authors for the revised manuscript and the given answers. All questions have been addressed.

Reviewer #4

(Remarks to the Author)

Point-by-point response to the reviewer comments

Reviewer #1 (Remarks to the Author):

Balancing the efficacy and safety of Chimeric Antigen Receptor T cell therapy by affinity combination

In this manuscript Warmuth, Dotsch, et.al, propose the novel concept that a combination therapy using anti-CD19 CARs with different affinities (2.7 nM and 140 nM) could achieve comparable efficacy to high affinity CARs alone, while lowering off-tumor toxicity. The authors suggest this strategy has the potential to lower the risk of cytokine release syndrome (CRS) and immune effector cell-associated neurotoxicity syndrome (ICANS), thereby expanding the therapeutic window of CAR T cells. The authors support their hypothesis through a series of in vitro and in vivo assays, including studies in NSG and a humanized mouse model, to assess both efficacy and toxicity. A particularly notable finding is that high affinity CARs in the bone marrow exhibit a distinct phenotype in the presence of low affinity CARs as determined by scRNA-seq. The authors suggest a synergistic interaction between the low- and high-affinity CARs, enabling a balance between therapeutic efficacy and safety.

The work is relevant to the field of CAR T cell immunotherapy, where achieving an optimal balance between efficacy and toxicity remains a major challenge. The proposed affinity-combination approach should be of interest to the immunotherapy community. However, the scope of the study is currently limited to CD19-directed CARs with a specific architecture (a CD28 transmembrane domain, a 41BB co-stimulatory domain and a CD3z signaling domain) which may constrain the broader applicability of the findings. While it is reasonable for the authors to stay within this defined system, expanding the study to include additional target antigens and/or other CAR architectures would substantially increase the impact of the approach.

We thank the reviewer for emphasizing the relevance of our study and for the valuable suggestion to extend our findings to additional target antigens. Indeed, this recommendation and consequent experiments clearly increased the impact of our findings. To address this point, we evaluated an additional pair of high- and low-affinity CARs recognizing the same epitope on the human ROR1 antigen (high-affinity CAR R12 - $K_D = 0.11$ nM; lower-affinity CAR R12v20 - $K_D = 0.98$ nM) (Hudecek et al., Clin Cancer Res. 2013). For this CAR pair, we assessed cytotoxic activity and the expression of exhaustion-related markers following chronic stimulation, using CAR-T cells either alone or as affinity-combined products.

High-affinity R12-T cells (R12^{hi}-T) displayed superior tumor cell killing compared with lower-affinity R12v20-T cells (R12^{lo}-T). Notably, a 50:50 mixture of high- and low-affinity CAR-T cells (corresponding to a 0.5x dose of each CAR) achieved cytotoxic activity comparable to that of R12^{hi}-T cells used alone at full dose (1x). Moreover, the affinity-combined product outperformed both R12^{lo}-T cells at full dose and R12^{hi}-T cells at half dose, which corresponds to the high-affinity CAR-T cell dose present in the combined product. In chronic stimulation assays, R12^{hi}-T cells exhibited higher expression of multiple exhaustion markers compared with R12^{lo}-T cells, whereas the affinity-combined product displayed exhaustion levels comparable to those of the lower-affinity CAR-T cells. Owing to the lack of a reporter gene in the anti-ROR1 CAR constructs, it was not possible to discriminate between the two CAR-T cell populations within the combined product.

In summary, the ROR1 antigen model corroborates the observations made in the CD19 model, demonstrating improved functionality and less signs of exhaustion of affinity-combined CAR-T products compared with high-affinity CAR-T cells alone also for another antigen model. These findings further underline the significance of CAR mixing, and were added in **Fig. S11**.

The authors present affinity measurements for a well-established high affinity murine ScFv (JCAR017 based on FMC63 - 2.7 nM) and a lower affinity humanized version (JCAR 021 - 140 nM). They then developed a collection of affinity variants by mutating the JCAR021 scFv and incorporating these variants into synthetic CAR constructs to evaluate performance. While this provides a careful and systematic evaluation of affinity-dependent CAR function, similar studies have been previously reported. For example:

He et. al (Science Immunology, 2023) characterized several affinity variants of FMC63, including WT - 4.5 nM, Y261A - 682.5 nM, Y70A - 275 nM and showed functional differences in CAR T cells depending on affinity.

<https://www.science.org/doi/10.1126/sciimmunol.adf1426>

Barden et al (Journal for Immunotherapy of Cancer, 2024) and Liu et al have explored the relationship between ScFv affinity and CAR efficacy in other tumor antigens, such as HER2, with conclusions that moderate affinity cAR may outperform both high and low affinity variants under certain conditions.

<https://jitc.bmj.com/content/12/12/e010208>

Given this context, the novelty of the current work lies not in the generation of affinity variants but in the combination of two CARs of differing affinity to enhance the therapeutic index. This concept is interesting and underexplored.

We agree with the reviewer that the novelty of our work lies in the combined-affinity approach to improve the balance between efficacy and safety of CAR-T cell products. In order to follow the reviewer's advice, we have revised the Abstract, Introduction, and Discussion to emphasize this particular point. We believe that these changes, together with the addition of the suggested references, should clarify how our study corroborates prior work on affinity-dependent CAR function while providing a distinct contribution to the design of innovative and more efficient CAR-T cell therapies.

However, to strengthen the affinity exploration, the authors might consider including *in vitro* killing assays using CD19+ targets of different densities at lower E:T. Previous work has shown *in vitro* differences that are affinity dependent (Fig. 4C - reference above). Under the current conditions, where T cell numbers are high, such differences might be masked.

The overall conclusions of the manuscript are promising. However, several key claims, particularly those regarding the mechanism of synergy or communication between high- and low-affinity CAR T cells in the combined product, would benefit from additional clarification or supporting evidence.

We followed the reviewer's recommendation and performed additional *in vitro* killing assays using lower E:T ratios. These data are discussed below in the point-by-point response. We also tried to further elucidate potential mechanisms explaining the synergy, focusing on the hypothesis that epitope masking by lower-affinity CAR-T cells protects high-affinity CAR-T cell function. To this end, we attempted to visualize interactions between effector and target cells in *in vitro* co-culture systems using antibody blocking approaches, membrane staining/dye transfer as well as high-resolution microscopy. However, these assays proved technically challenging and did not provide a better assessment of the mechanism of action, which remains beyond the scope of this manuscript.

We revised the manuscript to ensure a usage of the term "synergistic" cautiously and only as a potential interpretation of the data, not as a firm conclusion.

Specifically – Major points

1. Efficiency of *in vitro* killing across a broad affinity spectrum of CAR binders

Current E:T ratios used to explore differential *in vitro* killing across variants (Fig. 2g) show early saturation in target killing for most variants. Most variants that exhibit intermediate behavior between the control curve and JCAR021 express low levels of CAR or do not express it at all (Suppl. Fig. 3d). More challenging E:T ratios may be able to demonstrate dependence of killing dynamics on CAR affinity. Similarly, exploring how variants with different affinities kill targets with low antigen density would provide another way to test differences *in vitro*. The authors already demonstrated a relationship between CAR affinity and capacity to kill low antigen density targets (Fig. 5n). Exploring affinity variants across these two axes (total antigen by E:T ratio or using low antigen density per cell) could help validate the claim that there are no functional differences across the broad spectrum of CAR affinities.

We followed the reviewer's suggestion and tested more challenging E:T ratios across CARs spanning the full affinity spectrum in our dataset, using the same CD19⁺ tumor cell line engineered to express either low or high target density. We did not observe improved resolution with respect to the dependence of *in vitro* target cell killing on CAR affinity and antigen density, except for the ultra-low, TCR-like CAR affinities. These new observations confirmed the reduced cytotoxic capacity of lower-affinity CARs, particularly against low-antigen-density target cells, already observed in Fig. 5n of the original manuscript. These data have been incorporated into the new **Fig. S4** of the revised manuscript.

2. CAR T cell toxicity associated with affinity.

Despite generating several affinity variants, the authors only explored the original 2 CARs based on JCAR017 and JCAR021. Although there is a clear correlation between CAR affinity, dose, and toxicity, it would have been interesting to observe variants with intermediate affinities between these two (e.g., S168C, S227K, S50A). This is particularly relevant because Fig. 4b and 4h suggest that for JCAR017, a dose between 1M and 4M is sufficient to control tumors without significant toxicity; however, not even 8M JCAR021 can control tumors.

We thank the reviewer for this constructive suggestion. We agree that inclusion of additional variants strengthens the interpretation of affinity-dependent effects on toxicity. In response, we analyzed the affinity variants S168C and S227K, which exhibit the highest affinities in our dataset while remaining substantially closer in affinity to JCAR021 than to JCAR017. T cells engineered with these CARs were evaluated using the *in vitro* monocyte-induced cytokine release (CRS) assay. Both S168C- and S227K-CAR T cells produced significantly lower levels of CRS-related cytokines compared with JCAR017-CAR T cells, with cytokine profiles closely matching those observed for JCAR021-CAR T cells.

In addition, we evaluated 1:1 mixtures of high-affinity JCAR017-CAR T cells with each of the newly analyzed lower-affinity variants. These mixed products were compared with two JCAR017-CAR T-cell controls: one matched for the total CAR-T cell dose and one matched for the number of JCAR017-CAR T cells present in the affinity-combined product. For both S168C- and S227K-CAR T cells, the mixed products induced significantly lower levels of CRS-associated cytokines than the same total dose of JCAR017-CAR T cells alone and exhibited cytokine release comparable to the half-dose JCAR017 control. This indicates that adding lower-affinity CAR-T cells does not increase toxicity beyond what is expected from reducing high-affinity CAR T-cell numbers.

Together, these data further support the affinity-toxicity relationship previously described in the manuscript for JCAR017 and JCAR021. The new findings have been incorporated into the revised manuscript as **Fig. 5a–e**.

3. Ratio of low/high affinity CAR T cells over time *in vivo* vs *in vitro* –

The dynamics of CAR T cell populations in the 50:50 combination differ substantially between the *in vitro* and *in vivo* settings. *In vitro* (Fig. 5g–i), the total expansion resembles the high-affinity CAR alone, but the population eventually shifts toward low-affinity CAR dominance (~30:70 by day 28,

Fig. 5i). In contrast, *in vivo* data from day 10 post-ACT (Supp. Fig. 12c, left) show similar cell numbers for both the low-affinity and combined products across different dose levels, which suggests different expansion or persistence dynamics. It would strengthen the manuscript if the authors could comment on these discrepancies and discuss possible biological or experimental reasons behind them.

We thank the reviewer for highlighting this difference and giving us the opportunity to elaborate on these population dynamics. A major difference between the *in vitro* and *in vivo* settings lies in the distinct antigenic pressures experienced by CAR-T cells. In the *in vitro* chronic stimulation model, target cells are repetitively added to the CAR-T cells, which are continuously and directly exposed to high levels of their cognate antigen. This drives constant expansion and proliferation without reaching full exhaustion. Consequently, both high- and low-affinity CAR-T cells maintain proliferation (**Fig. 5g-h**). In contrast, in the *in vivo* model, antigen exposure changes depending on the efficacy of the infused CAR-T cells. In mice treated with lower-affinity JCAR021-T cells alone, tumors continue to grow due to poor efficacy, providing persistent antigen stimulation that sustains T-cell expansion. Conversely, in mice treated with high-affinity JCAR017-T cells alone or in combination, tumors regress markedly, leading to diminished T-cell expansion. The small residual tumor burden maintains JCAR017-T cells at a steady level rather than inducing the typical contraction usually seen in the peripheral circulation with complete tumor clearance (**Fig. 6c and Fig. S13c**). What remains unclear is the exact underlying reason for the late expansion of the JCAR021-T cells also in the combined product. Future investigations will be necessary to unravel underlying mechanistic insights for those proliferation dynamics; we deliberately chose not to include speculative interpretations in the manuscript.

4. The authors propose that the high-affinity CAR T cells are protected from exhaustion in the combination setting. However, the absolute numbers of high-affinity CAR T cells (JCAR017) in both the solo and combination treatments appear to decline similarly over time (Supp. Fig. 12c, middle). Would earlier time points for the high-affinity CAR alone show a more comparable phenotype to that observed in the combination setting? Alternatively, could the dominant expansion of the low-affinity CAR population reduce the number of high-affinity CAR interactions with target cells, thereby indirectly preserving their phenotype?

The transcriptomic analyses in **Fig. 6g-j** clearly indicate that JCAR017-T cells display milder signs of exhaustion in the combined product. This is further supported by the assessment of a smaller set of exhaustion-related markers in **Fig. 5i-j** after chronic antigen exposure, demonstrating a consistent phenotype across assays. In contrast, expansion and persistence data do not provide insight into the functional state of the transferred cells and therefore cannot be used as surrogate markers of exhaustion. We also do not consider it necessary to determine whether high-affinity CAR-T cells alone would display a phenotype similar to the combination setting if analyzed at earlier time-points. In our view, the key and most informative finding is precisely that, under identical conditions, high-affinity CAR-T cells behave differently when administered alone versus in combination with low-affinity CAR-T cells.

As noted in the Discussion, we also considered the hypothesis that competition for antigen binding may be a central mechanism through which low-affinity CAR-T cells help preserve a more functional and less exhausted phenotype in the high-affinity CAR-T cell population, given recognition of the same epitope. Future experiments are required to gain more detailed insights into the exact consequences and dynamics of epitope competition between different CAR-T cell populations (we started first live cell *in vitro* imaging studies to visualize and compare target contact dynamics; although the data are highly interesting, we believe they are too preliminary to draw sound conclusions at this stage and decided not to add them to the manuscript).

5. It would be useful if authors split the analysis of the combined JCAR021 and JCAR017 CARs for each individual treatment group (0.8/0.8 and 0.2/1.4). These two groups have different tumor control profiles (Fig. 6c) and total number of cells (Suppl. Fig. 12d). It would be interesting to see if the described phenotypical differences are maintained when comparing JCAR017 alone with JCAR017 in the 50:50 ratio.

We thank the reviewer for these insightful suggestions, which substantially strengthened the interpretation of the observed phenotype. We have now separated the analysis of the combined JCAR017 and JCAR021 treatments into the individual 50:50 and 10:90 dosing groups. This stratification confirms a more pronounced exhausted phenotype in high-affinity JCAR017-T cells when administered alone compared with the affinity-combined conditions. Moreover, we observed that the shift toward a less exhausted phenotype in high-affinity JCAR017-T cells became progressively more pronounced when moving from the 10:90 to the 50:50 mixing ratio, with transcriptional profiles increasingly resembling those of low-affinity JCAR012-T cells. Notably, the 50:50 group exhibited a markedly reduced exhaustion phenotype, characterized by significantly lower expression of exhaustion-associated genes (e.g., TOX and CCL5), higher expression of genes linked to cytotoxic function (e.g., KLRD1, SCML4, and GZMB), and overall reduced exhaustion and dysfunction scores. We hypothesize that these differences reflect increased antigen pressure on JCAR017-T cells when delivered at lower doses, a notion further supported by the higher residual tumor burden observed under these conditions (Fig. 6c). These new analyses have been incorporated into the revised manuscript as Fig. 6h-j.

6. The current data set lacks an additional control using high-affinity CAR T cells with untransduced T cells or CARs of unrelated specificity at matched total cell doses. This limits the ability to conclusively attribute phenotypic shifts in the high-affinity CAR population to active synergy or indirect effects from the low-affinity CAR T cells. Including such controls would help determine whether the observed effects are specific to affinity combinations or merely a consequence of changes in T cell dose or competition.

We thank the reviewer for this suggestion. While we fully appreciate the relevance of the proposed control groups, we believe that we have tested a sufficiently broad and diverse set of experimental conditions that consistently converged on similar phenotypes indicative of a beneficial effect of the affinity-combined products. Overall, our data indicate that the affinity-combined CAR T-cell product can achieve efficient antitumor activity together with an improved safety profile, while preserving a more effector-like rather than exhausted phenotype.

The *in vivo* data shown in Fig. 6 suggest a potential synergistic interaction for the following reason. First, the 50:50 affinity-combined product exhibits significantly superior antitumor activity compared with JCAR017-T cells alone (Fig. 6c), despite lower expansion of JCAR017-T cells within the combined product (Fig. S13c). In other words, the improved antitumor efficacy exceeds what would be expected solely from an increased overall CAR T-cell dose (JCAR017 + JCAR021). Second, the milder signs of exhaustion of JCAR017-T cells in the combined product (Fig. 6g-j) were not readily anticipated.

Nevertheless, we acknowledge that these observations are not sufficient to conclusively demonstrate synergism. Accordingly, we have revised the manuscript and used the term “synergistic” cautiously and only as a potential interpretation of the data, not as a firm conclusion.

7. It remains unclear whether the differences in CAR T cell phenotype (especially exhaustion profiles) could be influenced by differential cytokine production. Is there *in vivo* data on cytokine expression profiles comparing high- and low-affinity CARs, or in the combination product? This could help determine whether secreted factors from the low-affinity CAR T cells contribute to shaping the phenotype or function of their high-affinity counterparts.

We appreciate the reviewer interest in cytokine profiling. Circulating cytokine levels were measured in a humanized mouse model of CRS using a cytokine panel; however, cytokine measurements were not performed in the tumor-bearing NSG mice, which were instead used for extended evaluation of CAR-T cell functionality, persistence, and phenotype. Importantly, the restricted cytokine panel available would not have been sufficient to support mechanistic conclusions regarding how low-affinity CAR-T cells modulate high-affinity CAR-T cells.

We also considered interrogating our scRNA-seq dataset for cytokine expression. However, in addition to the (experiment intrinsic) absence of a JCAR021-T-cell-alone control arm, cytokine expression is only moderately reliable at the single-gene, single-cell level in scRNA-seq data. Cytokine transcripts are typically low in abundance and exhibit rapid and transient transcriptional dynamics, resulting in sparse, noisy, and timing-dependent detection. While these limitations can sometimes be mitigated by aggregating large numbers of cells within the same cluster, our dataset did not meet this requirement. Moreover, even under optimal conditions, transcriptional cytokine signals are generally expected to be supported by complementary protein-level measurements. Although we agree that cytokine production is an important mechanism by which CAR-T cells may mutually modulate each other in affinity-combined products, the available datasets do not provide sufficient resolution or coverage to allow robust and mechanistically sound conclusions. Therefore, we chose not to include cytokine analyses to avoid overinterpretation.

Several issues could be clarified and revised to improve data interpretation. Addressing the points below would enhance clarity

- Replacing the labels “JCAR017” with “High affinity CAR (ScFv - 2.7 nM)” and “JCAR 021” with “Low affinity CAR (ScFv 140 nM)” throughout the figures would make the data more accessible and easier to follow for a broader readership, particularly for those less familiar with the names/codes.

We thank the reviewer for this suggestion. We have incorporated the affinity values (K_D) into the Fig.1 panels to provide additional clarity. However, we have retained the original CAR names (JCAR017 and JCAR021) in the other figures, as these are the established terms used throughout the manuscript and the field. This combination should allow the data to remain accessible while maintaining consistency with literature.

- CAT CAR T cells are only explored in Fig. 1. Although it is important that the authors benchmark their findings against other previously described CARs, this could be moved to the supplementary figures, allowing the main figures to focus only on the high and low affinity CARs that are explored further (JCAR017 and JCAR021).

We thank the reviewer for this suggestion but respectfully disagree with it. CAT CAR, as an effective low-affinity CAR in a similar affinity range as JCAR021, provide an important benchmark for comparison in Fig. 1. Including CAT CAR-T cells in the main figure allows readers to directly assess the relative effects of the high- and low-affinity CARs (JCAR017 and JCAR021) within the same experimental context. Moving CAT to the supplementary figures would reduce the clarity of this comparative assessment.

- Fig.1g axis label reads E:T 1:4 but the Methods section for the assay description states E:T 1:0.25, which one is it? Intuitively, one would expect a higher fraction of cells secreting cytokines at higher target numbers. The current axis labels suggest otherwise. Please, clarify

We thank the reviewer for bringing this to our attention. The correct effector-to-target ratio used in the assay is E:T 1:0.25, as stated in the Methods section. The axis label in Fig. 1g was inadvertently mislabeled as “1:4” during figure preparation; mathematically, E:T 1:0.25 corresponds to 4:1, and this inversion led to the incorrect annotation. We have now corrected the axis label in Fig. 1g to accurately reflect the E:T = 1:0.25 ratio and ensure consistency with the Methods description.

- Fig. 1h More E:T conditions and antigen-densities for each construct could reveal differences in the *in vitro* killing behavior, particularly under suboptimal T cell conditions where functional differences are more likely to emerge (see comment in previous section)

We assessed *in vitro* cytotoxicity across a range of E:T ratios down to 0.125:1 using a flow-cytometry-based co-culture assay with the CD19-expressing target cell line Nalm6 engineered to express either wild-type or low levels of CD19. Because Nalm6 cells do not adhere optimally and are therefore unsuitable for xCELLigence-based killing assays, we selected a flow-cytometry-based approach, which also enables higher-throughput analysis of multiple conditions in parallel. While reduced antigen density and lower E:T ratios progressively affected the magnitude and efficiency of cytotoxicity after 24 hours, both JCAR017 and JCAR021 exhibited a comparable, parallel decline in target-specific killing against wild-type and CD19-low Nalm6 cells. These data have been incorporated into the revised manuscript as **Fig. S1d–e**.

- On page 6, the phrase "the individual label of CAR-T cells with fluorescence conjugated anti-CD45 antibodies enabled multiplexing..." seems to indicate that CAR-T cells were generated and stained, while the methods section states that target B cells were stained with anti-CD45 antibodies. Please clarify this discrepancy.

We thank the reviewers for pointing this out. Indeed, it is the target B cells that are specifically labelled with fluorescence-conjugated anti-CD45 antibodies to enable multiplexed k_{off} -rate measurements of 16 different scFvs simultaneously. We adjusted the phrasing in the Results section of the revised manuscript.

- The logic of Fig. 2 is currently somewhat confusing, as it first addresses and classifies CAR T cells based on function, and only later explains that many of the low-functioning CARs do not even express the receptor as a surface protein. One suggestion for reorganizing the figure is to first establish the affinity of the variants, then describe their expression as CARs in primary T cells, and only proceed with the functional assays for those that show clear CAR expression (based on EGFRt+ and STII+). Additionally, binning the CARs into groups does not allow the reader to appreciate interesting cases of function versus affinity. The heat map in Fig. 2i is a good summary, but including some XY plots of affinity versus target lysis or affinity versus IL-2 secretion could help to highlight interesting cases. That would also help to feature variants that are used in following figures, allowing a better interpretation of those results.

We thank the reviewer for the helpful suggestions regarding the structure and clarity of Figure 2. In line with this feedback, we have reorganized the figure to present the data in a more intuitive sequence. After the affinity measurements, the revised figure now continues with an assessment of CAR surface expression in primary T cells, followed by the functional evaluation of the variants. For the functional analyses, we retained the mutants with suboptimal CAR expression for completeness, as impaired *in vitro* functionality in our dataset is predominantly linked to reduced receptor surface expression rather than affinity. This observation underscores the importance of considering receptor expression quality when interpreting the relationship between functionality and affinity, and further supports grouping CAR variants based on surface expression, as it represents the main source of functional variability in *in vitro* assays.

In addition, we performed the suggested XY analyses (see below, k_{off} -rate vs. surface expression, k_{off} -rate vs. target lysis, k_{off} -rate vs. IL-2 secretion). However, these did not yield additional mechanistic insights or highlight specific outlier variants, presumably due to the dichotomous nature of the functional differences rather than a continuum of variability within our dataset. Therefore, we provided these analyses to the reviewers but decided not to incorporate these data into the revised manuscript.

Together, these adjustments clarify the logic of Figure 2 and improve alignment between the characterization of CAR variants and the downstream experimental design. Corresponding updates have been made to the Results section.

Pearson correlation analyses examining the relationship between scFv k_{off} -rates and CAR surface expression (a), cytotoxic capacity determined as target lysis following 24 h co-culture with Raji tumor cells (b), and intracellular IL-2 production (c) in CAR-T cells. JCAR021 (light blue), CAR^{mfunc} variants (green) or $CAR^{lowfunc}$ variants (yellow).

- Fig.4 can be rearranged to improve clarity. Figure panels that showed immune reconstitution (Suppl Fig.7 c and d) could be moved to main Fig.4. Also, dose titration is interesting mostly only in the context of affinity variants. Panels Fig.4 b to g can be moved to supplementary, allowing a clearer focus to Panels Fig.4h to k. Some panels from Suppl Fig 9 can be then moved also to the main Fig. 4.

We appreciate the reviewer's thoughtful suggestion to reorganize Fig. 4 for improved clarity. Accordingly, we have focused Fig. 4 on the assessment of CAR affinities and varying doses in relation to CRS, while relocating the dose-titration experiment of high-affinity JCAR017-T cells and the initial comparison of high- and low-affinity CAR-T cells to the Supplementary Figures (Fig. S8 and S9, respectively).

- Which cell line is used in Fig.4 experiments (Raji or Nalm-6), please specify in figure's legend For each figure (Fig. 4, Fig. S8 and Fig. S9), we have added the corresponding experimental schematic, including the target tumor cell line used.

- Mapping the mutations in the JCAR021 ScFv onto its structure and color-coding them by functional impact would add valuable insight. This visualization could highlight whether key residues are localized near the CD19 binding interface or whether functional "hotspots" exist in certain regions of the antibody. This may also help clarify why some affinity variants behave differently than expected.

We appreciate the reviewer suggestion; however, mapping the mutations within the JCAR021 scFv would be informative only if the binding interface between JCAR021 and the CD19 epitope were structurally resolved. To the best of our knowledge, neither the residues of the JCAR021 scFv that directly interact with CD19 nor the crystal structure of JCAR021 have been determined to date. In contrast, the binding interface between JCAR017 (FMC63 clone) and CD19 has been structurally resolved (He et al., Sci Immunol 2023). Moreover, our data indicate that JCAR019 and JCAR021 recognize the same CD19 epitope (Fig. 1d). Still, this information alone is insufficient to reliably infer the CD19-interacting residues of JCAR021, beyond approximate predictions based on sequence similarity to JCAR017 and other immunoglobulin-like interactions. Such similarity-based prediction is precisely the approach implemented in the ABodyBuilder software, which leverages existing structural knowledge of related protein-antigen complexes to propose single-amino acid substitutions predicted to reduce receptor affinity. Consistent with the inherent limitations of purely predictive algorithms, only a subset of these mutations in our dataset preserved CAR expression while successfully reducing affinity.

Moreover, the affinity-modulating mutations identified in our study are distributed across the entire variable domain, encompassing both complementarity-determining regions (CDRs) and framework regions, without revealing a clear empirical association between mutation location and

functional outcome. For example, CAR variants exhibiting the strongest affinity reductions resulted from mutations located both near the N-terminus (V48E, H38G) and near the C-terminus (V236W, V236F) of the primary amino acid sequence. Similarly, substitution of the same polar residue with different non-polar amino acids produced divergent outcomes, ranging from complete loss of CAR expression (Y174G) to only a modest reduction in affinity (Y174V).

In summary, while the reviewer suggestion is compelling and could indeed yield valuable insights into the relationship between mutation position and functional consequences, addressing this question rigorously would require dedicated structural studies, beginning with the resolution of the CAR–antigen interaction interface.

To partially address this point, we have highlighted in yellow residue substitutions that resulted in structural dysfunction in the existing Supplementary Table 1, which summarized the biochemical characteristics and positional context of all variable residues that were mutated.

- Fig. 5 would benefit from including data for all the variants mentioned in the figure. The monocyte cytokine release assay and the chronic stimulation assay should be performed with all the described variants, as well as their combinations. Additionally, at least one variant with intermediate affinity between JCAR017 and JCAR021 should be tested (e.g., S168C, S227K, S50A). This would allow for better interpretation of the role of affinity in toxicity and exhaustion prevention.

Again, we thank the reviewer for this constructive suggestion. As discussed above (point 3), we analyzed additional JCAR021-derived CAR variants with respect to toxicity, both as single products and in combination with high-affinity JCAR017-CAR T cells. These results have been included as **Fig. 5a-e** of the revised manuscript and are addressed in details in the corresponding section of this rebuttal (point 3).

Using the same receptors and experimental approach, we further extended the chronic stimulation assay to additional lower-affinity CARs, specifically the S168C and S227K mutants. T cells expressing these lower-affinity CARs behaved similarly to JCAR021-CAR T cells. Specifically, S168C- and S227K-CAR T cells exhibited comparable expansion kinetics under chronic stimulation, regardless of the presence of high-affinity JCAR017-CAR T cells, and consequently became the dominant population within the combined cell product, as also observed for JCAR021-CAR T cells. Moreover, S168C- and S227K-CAR T cells expressed lower levels of the exhaustion markers PD-1, LAG-3, and TIM-3 when cultured alone and contributed to reduced expression of these markers in JCAR017-CAR T cells within the affinity-combined product. These data have been incorporated into the revised manuscript as **Figure 5f-j**.

- Similarly, antigen density is an important factor to evaluate, as it may reveal differences in target killing by the various affinity variants, as observed in Fig. 5n. In addition, Fig. 5n shows that target killing by the combination of CAR T cells appears to reach saturation at 24 hours, which does not allow for a proper evaluation of whether the effect of the combined CAR T cell approach is driven by phenotypic differences in the high-affinity CAR. Testing different E:T ratios could help the reader observe these potential differences, such as instances where the same amount of high-affinity CAR fails to eliminate tumor cells while the combination, even at a lower high-affinity CAR dose, is effective. Furthermore, testing similar ratios of JCAR017 and JCAR021 alongside the other variants would also be informative.

As already discussed before (point 1), we tested more challenging E:T ratios across CARs spanning the full affinity spectrum in our dataset, using low and high antigen-expressing CD19⁺ tumor cells. We did not observe improved resolution with respect to the dependence of *in vitro* target cell killing on CAR affinity and antigen density, except for the ultra-low, TCR-like CAR affinities (**Fig. S4**).

We therefore focused on the two ultra-low-affinity variants (V236W and V236F) that exhibited impaired *in vitro* cytotoxicity under conditions of high E:T ratios and low antigen density and combined them at a 1:1 ratio with high-affinity JCAR017 CAR T cells. As in the *in vitro* CRS model,

the mixed products were compared with two JCAR017-CAR T-cell control conditions matched either for the total CAR T-cell dose or for the number of JCAR017-CAR T cells present in the affinity-combined product. Notably, the combined products rescued the suboptimal cytotoxic capacity observed with low-affinity CAR T cells alone, achieving levels of target cell killing that exceeded those of the 0.5x dose JCAR017 control, particularly under stringent conditions of high E:T ratios and low antigen density. These data have been incorporated into the revised manuscript as **Fig. S10**.

Figure 6e. Why does there seem to be much less cells for both high- and low-affinity CAR T a in the UMAP in the 50:50 condition than in the 90:10 condition? The absolute counts in S12c show otherwise, could you provide an explanation between the count trends in the blood and bone marrow? Similarly, as described before it would be useful if authors split the analysis of the combined JCAR021 and JCAR017 CARs for each individual treatment group (0.8/0.8 and 0.2/1.4). Also, including it would be useful for the interpretation of the results to include in the comparisons in Fig. 6h an j the profile of JCAR021 alone and in combination.

The UMAPs are derived from scRNA-seq analyses of bone marrow aspirates and should therefore be compared with the data shown in **Fig. 6d** rather than **Fig. S13c** (revised manuscript), which reflects peripheral blood samples. The total cell counts in Fig. 6d indeed correlate with the relative abundances inferred from the UMAPs. In addition, technical factors related to sample preparation and data processing should be considered, as these may account for the apparently lower representation of JCAR017-T cells in the UMAPs under the 50:50 condition.

Mice treated with JCAR021-T cells alone had to be euthanized due to excessive tumor burden prior to the scRNA-seq time point (**Fig. 6c**). Consequently, this key control was not available, precluding differential gene expression analyses analogous to those performed for JCAR017-T cells in **Fig. 6f**. Nevertheless, JCAR021-T cells from the combined groups were included in the analysis of exhaustion-associated gene expression (**Fig. 6g**), which revealed no evidence of exhaustion or functional impairment.

The authors claim: “Our data show that lower-affinity CAR-T cells remain functionally active in the presence of high-affinity CAR-T cells, leading to a more controlled and sustained immune response.” However, this claim is difficult to support with the current experiments. Although it can be assumed that lower quantities of JCAR017 require the presence of JCAR021 to achieve similar survival (Fig. 6b), effective tumor control appears to require at least 0.8 million high-affinity CAR-T cells to prevent tumor growth (Fig.6c). Additionally, low-affinity CAR-T cells achieve similar numbers in the bone marrow under conditions that differ in terms of tumor control (i.e., 1.6 million single low-affinity cells versus 0.8 million in combination) (Fig.6d). Including an analysis of JCAR021 from the scRNAseq data, as previously suggested, may help to substantiate this claim. We thank the reviewer for pointing this out. Indeed, the sentence contained an error resulting from an inversion of the terms “lower” and “high.” The corrected sentence now reads: “Our data shows that high-affinity CAR-T cells remain functionally active in the presence of lower-affinity CAR-T cells, leading to a more controlled and sustained immune response”. The manuscript has been corrected accordingly.

Please clarify if the same donor T cells were used throughout the manuscript or different ones. If a single one, adding other donor T cells would strengthen the results.

We appreciate the reviewer attention to donor variability. To clarify, different primary T-cell donors were used for each independent mouse experiment. This approach was intentionally chosen to ensure that *in vivo* outcomes were not dependent on a single donor’s T-cell biology. In cases where data from multiple experiments were pooled, the use of different donors is explicitly indicated in the corresponding figure legends. We agree that donor diversity strengthens the robustness of the findings, and the current dataset already reflects this by incorporating multiple independent donors across experiments.

Methods are mostly adequate. Extra details in some sections would help to reproduce authors' work. Specifically:

- Add source of retroviral expression vector (Addgene?, other lab?)

Source is now included.

- Which number of RD114 that was seeded, in which plate/flask format, for how long before transfection.

This information is now included.

- In which media transfection was performed, and was media exchanged during those 72h?

This information is now included.

- For how long the transduced primary T cells were maintained in culture before CAR expression was analyzed. How long after the transduction they were used in in vitro experiments. How long for in vivo experiments.

This information is now included.

- For which experiments CAR T cells were sorted? For which experiments they were not sorted. Please add this information to the figure legend, particularly for the in vivo experiments.

This information is now included.

- For the chronic restimulation assay, please describe the plate format used. Also, add how many rounds of stimulation were performed.

This information is now included.

- For BM and spleen preps, do authors intend to write ACK buffer instead of "ACT buffer"?

We thank the reviewer for pointing this out. In our manuscript, the term ACT buffer was used in some instances following historical usage in our laboratory protocols and German-language documentation. Functionally, ACT in our context corresponds to the RBC lysis buffer (ACK buffer) used for isolation of nucleated cells from blood or bone marrow.

Reviewer #2 (Remarks to the Author):

Warmuth et al. present interesting results showing that the combination of high- and low-affinity CARs targeting Cd19 antigen can balance the antitumor response by reducing toxicities without compromising efficacy. The authors generated and thoroughly tested a panel of CD19 binders and compared them to clinically used CAT. The authors evaluated T cells with different binders alone or in combination in vitro and in vivo, including in a humanized model to assess CRS. Overall, the manuscript is well presented, original, and addresses a critical question in the field. My only concern is that the study would benefit from deeper mechanistic analyses to determine how low-affinity CARs mitigate high-affinity CAR exhaustion and excessive stimulation. Is this achieved through transient antigen masking, reduced tumor burden, or potential cytokine secretion? Addressing this question would further strengthen the authors' conclusions. A minor point that should be addressed in the limitations section is the challenge of translating these findings to patients, given the heterogeneity both between patients and across different antigens, a limitation noted by the authors in the discussion.

We thank the reviewer for recognizing the relevance of our study and agree on the importance of deeper mechanistic analyses, which we attempted to address as outlined below.

First, to extend our findings, we evaluated an additional pair of high- and low-affinity CARs recognizing the same epitope on the human ROR1 antigen (high-affinity CAR R12 - $K_D = 0.11$ nM; lower-affinity CAR R12v20 - $K_D = 0.98$ nM) (Hudecek et al., Clin Cancer Res. 2013). Similar to the CD19 model, a 50:50 mix of high- and low-affinity CAR-T cells achieved cytotoxicity comparable to full-dose high-affinity R12-T cells while outperforming lower-affinity R12v20-T cells, and exhibited significantly lower exhaustion levels than high-affinity CAR-T cells after chronic stimulation (**Fig. S11c-e**).

Second, we tried to investigate deeper the hypothesis that epitope masking by lower-affinity CAR-T cells may contribute to the preservation of functionality in high-affinity CAR-T cells when both CARs engage the same epitope. We attempted to assess by live-cell imaging microscopy whether lower-affinity CAR-T cells interact more frequently with target cells - presumably due to shorter dwell times - thereby masking antigen availability for high-affinity CAR-T cells. We also tested reviewer's suggestion of receptor masking by using anti-CD19 antibody of the same clone like JCAR017. However, establishing appropriate experimental conditions for meaningful analyses proved technically very challenging.

Finally, we considered interrogating our scRNA-seq dataset for cytokine expression (**Fig.6**). However, in addition to the absence of a JCAR021-T-cell-alone control arm, cytokine expression is only moderately reliable at the single-gene, single-cell level in scRNA-seq data. Cytokine transcripts are typically low in abundance and exhibit rapid and transient transcriptional dynamics, resulting in sparse, noisy, and timing-dependent detection. While these limitations can sometimes be mitigated by aggregating large numbers of cells within the same cluster, our dataset did not meet this requirement. Moreover, even under optimal conditions, transcriptional cytokine signals are generally expected to be supported by complementary protein-level measurements.

In summary, the ROR1 antigen model corroborates the observations with the CD19 model, demonstrating improved functionality and less signs of exhaustion of affinity-combined CAR-T products compared with high-affinity CAR-T cells alone. However, further experiments will be required to elucidate the mechanisms underlying the beneficial effects of affinity-combined CAR-T cell products, which are beyond the scope of this manuscript.

Reviewer #3 (Remarks to the Author):

The presented manuscript elaborates the impact of receptor-affinity towards efficacy and safety of CD19 targeting CAR T cells. In short, the authors describe that *in vivo*, but not *in vitro* function correlates with affinity. Using improved murine CRS models, the authors show that in high, but not in low affinity CAR Ts, toxicity is correlated to dose. In a last step, the authors evaluate the combination of high and low affinity CAR T cells with the aim to maintain efficacy by reducing toxicities such as CRS.

For long time, CARs were preferentially generated from high affinity antibodies, whereas nowadays, scientists are becoming aware that sometimes, lower affinity might be beneficial in specific indications (e.g. Olson et al, Leukemia, 2022; Flugel et al, Nat Rev Clin Onc, 2023). Thus, the combinatory approach presented here is of significant interest to the field and displays a concept that has not been presented yet.

Most of the conclusions and claims are sufficiently supported, however, due to the large amount of data (varying dosages and combinations) in the murine studies, the reader would benefit from further explanation how studies are connected to each other, and if and how conclusions from one experiment might have shaped the setup of the following study. From the data presented, it looks like 2 Mio JCAR017 are typically curative in the used models, whereas 1Mio JCAR017 is not sufficient to completely eradicate the tumor. Regarding the combination of JCAR017 and JCAR021, there seem to be different ratios that might be beneficiary, and a final conclusion on the optimal ratio is missing. Thus, a final experiment is missing to fully support the main message of the manuscript: Within the CRS model, compare an optimal combination (e.g. 1Mio +1Mio, if that is the chosen dosage) against I) a comparable dose (total cell amount =2 Mio) of high affinity CAR and II) same amount of high affinity CAR as in the combination (1Mio) to make a final decision on the dose as well as to have supportive data on efficacy and toxicity, preferably over long term to address toxicity as well as long term efficacy and potential relapse.

We thank the reviewer for pointing out that our combinatorial CAR-T cell approach “is of significant interest to the field and displays a concept that has not been presented yet.” We understand the reviewer’s confusion and questions regarding the recommended cell dose for the affinity-combined product. They arise in part from the necessity of using different experimental models for distinct objectives. Some studies were performed using Nalm6 cells, whereas others employed the more aggressive Raji model. These models differ substantially in growth kinetics and immune evasion properties, which inherently influence the dose–response relationships observed across experiments. In addition, CAR-T cell doses used in humanized mouse studies cannot be directly translated to non-humanized xenograft settings. Humanized mouse models require higher cell doses to induce CRS and are not suitable for long-term monitoring due to xenogeneic and allogeneic immune interactions. Consequently, safety and functionality studies necessarily had to be uncoupled.

Despite these discrepancies, a consistent finding across models is the beneficial effect of a 50:50 combination of high- and low-affinity CAR-T cells compared with high-affinity CAR-T cells alone. This affinity-combined product mediated reduced CRS (*in vitro*), protection from exhaustion of high-affinity CAR-T cells, and improved anti-tumor activity (revised **Figs. 5 and 6**). Notably, for CRS evaluation - although performed *in vitro* - we incorporated the reviewer’s suggestion and directly compared (i) the proposed optimal 50:50 combination dose, (ii) an equivalent total dose of high-affinity CAR-T cells, and (iii) the corresponding single-dose high-affinity CAR-T cell condition.

To improve clarity, we have added a section to the study limitations describing the need of using different dosing regimens throughout the manuscript while explicitly identifying the 50:50 combined-product dose as optimal for further investigation.

Statistics should only be calculated if $n \geq 3$ and biological (but not technical) replicates are available. Smaller datasets can still be presented but should not be included in statistical analysis.

All *in vitro* data have been performed in at least three independent biological replicates, whereas mouse experiments include at least four mice per group (except Fig. S8), thereby fulfilling the statistical requirements.

Besides the above mentioned general comments, there are some details that should be addressed:

Introduction:

- Besides JCAR017 being the “central component of most commercially approved anti CD19 CAR T products” it should be mentioned that also JCAR021 is being evaluated in clinical trials and results have been reported (PMID 39820359 by Gauthier et al, 2025).

We thank the reviewer for bringing this to our attention. The study has been added to the Introduction.

Figures:

- Fig. 1h: The wording of figure should be revised: “data was obtained from one experiment”.

The text now reads: “Each dot represents a technical replicate ($n = 2-3$) of one out of three independent biological experiments”. The CAR-T cell cytotoxicity assay in Fig 1h was performed using three independent biological replicates; however, for simplicity, only one representative replicate is shown, as the same trend was observed across all biological replicates. We acknowledge that the reviewer’s comment may have arisen from a misinterpretation suggesting that only a single experiment had been performed.

- Fig 2f: How are constructs assigned to group mfunct and mfunct low? If this assignment is based on function, there is one construct within the CAR^{mlo} group that shows cytokine production after antigen spec. stimulation. Please clarify allocation.

The allocation of CAR mutants into functional groups was based on the efficacy of vector delivery and CAR surface expression, which we assessed via the transduction marker EGFR^t and the CAR expression marker STII, respectively. CAR^{mfun} preserved an optimal CAR surface expression, as indicated by the correlation between STII and EGFR^t highly comparable to JCAR017 and JCAR021. In contrast, CAR^{mlo} displayed either weak co-expression of the CAR and EGFR^t markers, or transduction rates below 20%. In the revised manuscript, we have reorganized Fig. 2 so that surface expression is presented as the first parameter, making the rationale for group allocation clearer.

- Fig. 2h: In addition to the transduction rate, is the MFI also reduced in the mlo group?

The surface expression profiles of CAR^{mfun} and CAR^{mlo} constructs are shown in Fig. S3a. Constructs lacking detectable STII on the cell surface consistently exhibit reduced MFI among successfully transduced cells. Moreover, CAR^{mlo} variants with overall lower gene delivery or integration efficiencies display correspondingly decreased STII MFI. As summarized in the accompanying figure, CAR^{mlo} constructs uniformly show lower MFI values compared with CAR^{mfun}, underscoring their reduced surface expression (see below).

CAR surface expression indicated as Mean Fluorescence Intensity (MFI) for CAR-T cells expressing JCAR021 (light blu), CAR^{mfunct} variants (green) or CAR^{lowfunct} variants (yellow).

- Fig 3f/S5b: BLI data from the one mouse surviving in the L237W group (d28) is shown in S5b, but not in 3f. Please justify why this has been excluded from the analysis.

We appreciate the reviewer's comment. In the original version, the data point from the single remaining mouse in the L237W group at day 28 was not included in Fig. 3f because this figure displays group-level summaries (mean + SD), which cannot be meaningfully calculated for n = 1. To avoid implying statistical representativeness where it is not present, this individual value was shown only in the BLI image in the supplementary figure. For completeness and transparency, we have now included the full dataset in the revised version of Fig. 3f.

- Fig. 3g ff: What is the baseline of the Mock T cells? Were any Mock T cells detectable in blood, BM or spleen?

Mock T cells were indeed detectable in blood (day 7), and bone marrow (BM) and spleen at the corresponding endpoints, as shown in the figure below (a = blood at day 7, b = BM and c = spleen at the endpoint). Mock T cells appeared at high levels during the early time course in the peripheral blood and in the spleen, but are barely detectable at the tumor site, which is expected given their lack of antigen specificity and thereby infiltration at the relevant tumor site.

Although mock T cells represent an important control for survival and residual tumor burden analyses, they provide limited information for assessing CAR-T cell persistence and phenotype at later time points, as mock-treated mice succumb to tumor progression within 10–14 days after tumor injection; accordingly, this group was not included in Fig. 3g.

Quantification of unedited (mock) or CAR-T cells adoptively transferred into tumor-bearing mice. Related to Fig.3 of the revised manuscript.

- Fig S6: Please include average radiance analysis of the remaining mice of d24 as the final day of the experiment

The average radiance values for the remaining mice at day 24 have now been included in the revised analysis and are shown in Fig. S7e of the revised manuscript.

- Fig. 4: How has the murine CRS model been improved? Please indicate for each experiment which cell line was actually used (Nalm-6 or Raji).

To improve the murine CRS model, we performed HSC titrations to generate larger cohorts of humanized mice derived from a single cord blood donor (Schütz et al., 2025 preprint). This approach reduces allo- and xeno-reactive responses by limiting the number of pooled HSC donors, which are commonly used in commercially available preparations. By minimizing donor heterogeneity, the model achieves more consistent engraftment and immune composition, improving reproducibility and reliability of the experiments. We added some of these advantages to the revised manuscript.

For clarity, the specific cell line used in each experiment (Nalm-6 or Raji) is now indicated in the respective figure panels.

- Fig 4h-k: Even though Fig 4 is meant to demonstrate toxicity, the BLI imaging suggests that the low affinity CAR might have an antitumor effect at high dosage while showing a lower toxicity. Is there data available on longer observation periods regarding efficacy?

Indeed, the BLI signals in Fig. 4h-k of the original manuscript (now **Fig. 4f-j**) indicate that low-affinity JCAR021-T cells exert antitumor activity while maintaining a more favorable toxicity profile. Consistently, we previously observed a degree of antitumor activity for JCAR021-T cells in immunocompromised tumor-bearing mice, as reflected by improved survival compared with untreated controls (**Fig. 3e-f**). However, this benefit is transient, as low-affinity CARs are unable to efficiently control tumor growth at later stages in our experimental setting. Unfortunately, the used *in vivo* CRS model has inherent limitations that prevent extended observation periods. Due to donor mismatches, xenogeneic responses and allo-reactive components, mice frequently develop rejection-related complications that constrain the feasible timeframe of analysis. Consequently, prolonged efficacy assessments beyond the time window shown in Fig. 4 cannot be reliably performed in this model, necessitating the uncoupling of toxicity evaluation from long-term efficacy following CAR-T cell adoptive transfer.

- Fig: 5b: Does the ametrine fluorescent protein influence CAR expression or functionality?

We appreciate the reviewer's question. In our experiment, Ametrine was co-expressed with the CAR and used *in vitro* and *in vivo* without any detectable impact on CAR surface expression. Moreover, Ametrine+ CAR-T cells behave as expected in functional assays, as shown in Fig. 5 and 6. Ametrine served solely as a fluorescent marker to track CAR-T cells.

- Fig 5c: Please clarify the composition of CAR017 and JCAR021 at t=0

At t=0 JCAR017 and JCAR021 cell numbers were counted separately and then combined at a 50:50 ratio to initiate the co-culture. After 24 hours, the composition remained largely stable with only minor changes in the relative proportion of the two CAR populations. Any deviations from the defined affinity combination were minimal, such that the initial configuration of the two products was essentially maintained.

- Fig 5a-e: From the presented data, it is not clear if the combination has a benefit at all. A control condition containing the reduced amount of JCAR017 without the addition of JCAR021 would be helpful to fully interpret the results.

To address this point, we updated **Fig. 5a-e** to include the reduced dose of JCAR017 alone, as suggested by the reviewer. In addition, we extended the analysis to include additional JCAR021-derived CAR variants and their corresponding affinity-combined products with JCAR017-T cells. Notably, a reduced dose of JCAR017-T cells (0.5x dose) induced stronger CRS than individual lower-affinity CARs administered at twice the dose (1x dose). More importantly, the 0.5x dose high-affinity CAR-T cells alone triggered, to some extent, higher cytokine release than the affinity-combined CAR products (0.5x JCAR017-T cells plus 0.5x lower-affinity CAR-T cells), indicating

that affinity combination does not exacerbate CRS despite the increased total cell number. These data have been incorporated into the revised manuscript as **Fig. 5a–e**.

- Fig.5n: please include a condition of 0.01×10^6 JCAR017 to enable a conclusion if similar lysis would also be achieved by the lower dose of JCAR017 in the complete absence of JCAR021. In the current data set, Figs. 5 and S10 do not sufficiently support the main message of the paper, as relevant controls are missing.

We agree with the reviewer on the importance of the suggested control, which was included in repetition experiments. Moreover, we extended the cytotoxic analyses to more stringent effector-to-target ratios and the use of CD19+ target cells with high and low antigen expression. Now, the mixed products were compared with two JCAR017-CAR T-cell control conditions matched either for the total CAR T-cell dose or for the number of JCAR017-CAR T cells present in the affinity-combined product. Notably, the combined products rescued the suboptimal cytotoxic capacity observed with low-affinity CAR T cells alone, achieving levels of target cell killing that exceeded those of the 0.5x dose JCAR017 control, particularly under stringent conditions of high E:T ratios and low antigen density. These data have been incorporated into the revised manuscript as **Fig. S10**.

- Fig. S10: same as Fig 5. In Figure S7, it is shown that giving 2 or 4 Mio of JCAR017 does not have a significant difference in efficacy. Thus, the data from S7 suggest that the observed effect is caused by the reduction of the amount of JCAR017, but not from the combination with the low affinity CAR.

We agree with the reviewer that the experimental setup of Fig. S10 in the original manuscript does not allow the observed reduction in tumor burden to be unequivocally attributed to the combined product rather than to the total dose of 2 million JCAR-T cells, as an additional control group receiving 2 million JCAR-T cells alone was missing. Accordingly, these data have been removed from the revised manuscript.

- Fig. S11: in contrast to the text (p.12), Fig. S11 doesn't demonstrate increased safety, but efficacy of combined CAR T. Indeed, a comparison of this "optimal combination approach" to standard dosages with regards to safety and efficacy would strongly support the manuscript.

We agree with the reviewer that Fig. S11 (now **Fig. S12** in the revised manuscript) demonstrates the anti-tumor efficacy of affinity-combined products rather than safety outcomes. This was indeed the intent of the experiment; however, we recognize that the original phrasing was suboptimal and led to misinterpretation.

In this experimental setting, we combined a fixed, suboptimal dose of high-affinity JCAR017-T cells - insufficient to achieve tumor clearance - with increasing doses of low-affinity JCAR021-T cells. The rationale was to assess whether higher doses of low-affinity CAR-T cells could rescue the suboptimal anti-tumor activity of high-affinity CAR-T cells. This strategy has potential implications for designing CAR-T products with an improved balance between efficacy and safety, as CRS severity correlates with the dose of high- but not low-affinity CAR-T cells (**Fig. 4**). However, we observed a beneficial anti-tumor effect of the combined product only at very high doses of JCAR021-T cells (16-fold excess), indicating that this approach is not optimal and would be difficult to translate into a combined product with an improved therapeutic index.

We have rephrased the rationale underlying this experiment accordingly and hope this clarification addresses the reviewer's comment.

- Fig. S11a: 1M JCAR017 and 1M JCAR017 +1M JCAR021 are not shown in b/c. if these conditions were not assessed here, but in a separate experiment (d/e), please delete in the experimental set up (a).

We thank the reviewer to point this out. We have removed 1M JCAR017 and 1M JCAR017 + 1M JCAR021 from the experimental setup in panel a (Fig. S12a of the revised manuscript).

- Fig. S11d/e is mentioned in the text in correspondence with Fig S12, maybe consider relocation. We understand the reviewer's point of view and agree that disrupting the numerical flow of figures and subpanels is not ideal. However, to strictly preserve the numbering sequence, Fig. S11d/e from the original manuscript would need to be placed as Fig. S13c (revised manuscript), which would disrupt the internal cohesiveness of Fig. S13. Therefore, although not optimal, we chose to retain the original figure order.

Discussion:

- The affinity variants of the CAR all target the same epitope within CD19. As one suggested mode of action is the competitive binding and availability of antigen binding sites, do you think the effects would be similar when targeting different epitopes?

We thank the reviewer for this insightful comment. We were also very interested in whether epitope specificity might influence the observed phenotype. To address this, we first evaluated an additional affinity-tuned CAR pair targeting the same epitope to confirm the superior features of affinity-combined CAR-T cell products over monospecific high-affinity CAR-T cells. Specifically, we analyzed CARs with variable affinities binding the same epitope on the human ROR1 antigen. Similar to the CD19 model, a 50:50 mix of high- and low-affinity CAR-T cells achieved cytotoxicity comparable to full-dose high-affinity CAR-T cells while outperforming lower-affinity CAR-T cells, and exhibited significantly lower exhaustion levels than high-affinity CAR-T cells after chronic stimulation. These data are presented in **Fig. S11** of the revised manuscript.

Moreover, we tried to investigate deeper the hypothesis that epitope masking by lower-affinity CAR-T cells may contribute to the preservation of functionality in high-affinity CAR-T cells when both CARs engage the same epitope. We attempted to assess by live-cell imaging microscopy whether lower-affinity CAR-T cells interact more frequently with target cells - presumably due to shorter dwell times - thereby masking antigen availability for high-affinity CAR-T cells. We further attempted to recapitulate the phenotype of affinity-combined CAR-T cell products by co-culturing high-affinity CAR-T cells with target cells in the presence of an anti-CD19 antibody of the same clone as that used in JCAR017. However, establishing appropriate experimental conditions for meaningful analyses proved technically very challenging. Further experiments will be required to elucidate the mechanisms underlying the beneficial effects of affinity-combined CAR-T cell products, which are beyond the scope of this manuscript.

Reviewer #4 (Remarks to the Author):

We thank the reviewer for their constructive feedback, which contributed significantly to the revision process and to improving the quality of the revised manuscript.